# Analysis and Forecast of the Use of E-Commerce in Enterprises of the European Union States

**Georgeta Soava** [1,*] **, Anca Mehedintu** [1] **and Mihaela Sterpu** [2]

1 Department of Statistics and Economic Informatics, University of Craiova, A.I. Cuza 13, 200585 Craiova, Romania; anca.mehedintu@edu.ucv.ro
2 Department of Mathematics, University of Craiova, A.I. Cuza 13, 200585 Craiova, Romania; msterpu@inf.ucv.ro
* Correspondence: georgeta.soava@edu.ucv.ro; Tel.: +40-251-411317

**Abstract:** This study analyzes the use of e-commerce by European Union enterprises. Based on an analysis of the evolution of the percentage of enterprises that perform e-commerce and the share of total turnover obtained from e-commerce, for the period of 2003–2020, a forecast for 2025 was made. These aspects were analyzed for 2015–2020 from the perspective of the Digital Economy and Society Index (DESI), highlighting the importance of e-commerce in the economic growth of each country by analyzing the share of gross domestic product (GDP) obtained from e-commerce in GDP. For studying the evolution of the indicators, a comparative analysis of the situation of individual and aggregate EU countries was performed, highlighting the evolution trends and determining the annual average growth rate indicator, used later for the short-term forecast. For the 2025 forecast, six regression models were used for the empirical estimation of the data. The results of the analysis show that the share of companies performing e-commerce and the turnover in e-sales vary significantly depending on the size of enterprises, and the results of the forecast estimate that by 2025 there will be a significant increase in e-commerce in most European countries.

**Keywords:** European Union; e-GDP; e-commerce; enterprises; DESI; selling online; turnover; forecast





## 1. Introduction

The digital economy, characterized by the ability to transform economies, jobs, and even society as a whole by introducing new technologies and digital processes, has made an important contribution to economic and social development. The digital economy encompasses transactions that result from countless online connections between people, companies, machines, and data.

In March 2021, the European Council set its vision for digital transformation by 2030 [1] and the digital goals that the European Union (EU) expects to achieve. At the same time, in order to align the DESI (Digital Economy and Society Index) with the cardinal points and targets corresponding to the Digital Compass, the structure of the DESI was restructured from five to four components. DESI includes thematic chapters on digital competitiveness in areas such as human capital, broadband connectivity, the integration of digital technologies by businesses, and digital public services, which help the member states to identify areas that need priority action [2].

In setting the objectives of the paper, we started from the context of the development of the economy and digital society in the European Union, and we focused on the third subcomponent of the DESI, e-commerce (EC), because it is one of the most important factors of the economic growth of a state. The "Digital Technology Integration" component covers two areas, "business digitization" and "e-commerce"; most of the studies focused on the first subcomponent and less on the second.

Small- and medium-sized enterprises (SMEs) now play a dominant role in almost all economies, whether they are developed or developing, contributing substantially to

the increase in national income. SMEs contribute significantly to each country's gross domestic product (GDP) [3]. One of the challenges of the new millennium facing SMEs is the digitalization and adoption of new technologies. In this regard, it is important to identify the level of adaptation of SMEs to digitalization [4], so companies must go through a coherent global process of digital transformation that provides opportunities for the expansion and internationalization of business [5,6], this being an important lever for ensuring economic growth [7–10].

Although digital transformations can ensure the development of companies [11], they can also be a challenge to SMEs in terms of their size and limited resources [12]; for this reason, the digitization strategies adopted at the level of SMEs aim at a shorter term compared with that of large companies [13].

At the EU level, the European Commission has seen the need for a single digital marketplace to create a space for cross-border e-commerce, where people can shop online from abroad and businesses can sell across the EU, regardless of where they are based in its territory [14].

In the digital economy, e-commerce has become an important technological and distribution tool, stimulating the competitiveness and innovation of companies in the way they promote their products and services, producing a transformation in human behavior, and producing a new consumer profile [15]. A number of authors demonstrate that digitalization is the driving force behind the growth of SME sales [16] by expanding e-commerce in global markets [17]. In addition, with the COVID-19 crisis, when physical interactions in business processes were limited, online commerce experienced a more pronounced development [18–21].

Based on these aspects, we considered it appropriate that this research focuses on the study of the evolution and forecast of the e-commerce component.

Following the analysis of the literature, it can be seen that there is little research that has paid attention to investigating the impact of e-commerce on GDP. At the same time, the analysis of the evolution of the number and type of enterprises that carry out e-commerce and of the turnover realized from this activity is quite little-approached in the specialized literature. In this regard, the objectives of this paper are:

1. Review of the literature on the digital economy and the impact of digitalization on companies, emphasizing the importance of e-commerce activity at enterprises level;
2. Economic analysis of the evolution of the e-commerce components of the DESI (the percentage of SMEs that are selling online; Selling, and the share of turnover obtained from e-commerce in the total turnover; Turnover);
3. Analysis of the evolution of the share of GDP from e-commerce in GDP (e-GDP);
4. Comparative analysis of the share of e-commerce enterprises in the total number of enterprises; Selling by types of enterprises (all enterprises; All, large enterprises; Large-, small-, and medium-sized enterprises; SMEs), and of the share of turnover in online sales in the total turnover; Turnover by types of enterprises (All, Large, SMEs);
5. Short-term forecast for DESI (Selling and Turnover) components and e-GDP;
6. Forecast for 2025 for the following components: selling for All, Large, and SMEs, and turnover for All, Large, and SMEs.

To this aim, we collected information for the period 2015–2020 for the indicators: e-commerce subcomponents of the DESI (selling and turnover) and e-GDP, and for the period 2003–2020, information for selling online and e-commerce turnover by type of enterprises.

The research is structured in two main directions. In the first part, we perform an economic analysis of the evolution of the DESI and e-GDP for the period 2015–2020 and a comparative analysis of the share of e-commerce enterprises in the total enterprises and of the share of turnover in online sales in the total turnover by types of enterprises (2003–2020). In the second part of the study, we perform forecasting for these indicators; we use the annual average growth rate (AAGR) to short forecast for 2021 the evolution of the variables from the DESI and e-GDP, and we use six different regression models (three

autoregressive models and three polynomial time regression models) aiming to forecast the other indicators for 2025.

The results of the study show that businesses, regardless of their size, are in a continuous process of adopting a new form of sales—e-commerce. In this sense, e-commerce can be an essential means for SMEs to expand into global markets. Thus, this research can contribute to the literature by completing the theoretical research on the analysis of companies that carry out e-commerce activity.

The results of the study could provide other researchers with important benchmarks for further research and could be the basis for action by the EU and national governments to achieve the objectives set by the digital policies in the Digital Agenda for Europe [1].

The structure of the paper is as follows: the second section presents a literature review; section three describes the models and methodology used; section four presents the results; and finally, section five presents a series of discussions and summarizes the main conclusions.

## 2. Literature Review

### 2.1. The Digital Economy

The definition of the digital economy has changed significantly since its first mention in the mid-1990s, due to the rapid development of technologies and their integration into various socio-economic activities [22]. The evolution of the concept of the digital economy has been based on research into various policies to support the introduction of digital technologies, on the one hand, and the increasing use of information and communications and digital technologies in business, on the other [23].

The challenges posed by the use of information and communication technologies in the global economy have become an important point of reference in EU policy and strategic orientation. To this end, in order to measure progress in the development of the digital economy and society, a Digital Economy and Society Index (DESI) [2] was developed and adopted in the European Union in 2014.

The DESI is an annual tool for analyzing and comparatively assessing the digital competitiveness of different individual economies, but also of the dynamics, both at the level of EU member states and at the level of groups of countries or across all the EU [24,25].

In this regard, a comparative analysis of digital performance with a sample of 45 countries (the 27 EU member states and 18 non-EU countries worldwide) permitted the identification of areas that need investment and the key actions needed to reach the levels of the best-performing countries, which could be the basis for monitoring the progress of the digitization of EU countries in the period 2021–2030 [1]. At the same time, the analysis of the relationship between the DESI and the sustainable development indicators demonstrates the significant impact of digitalization on sustainable economic and social development [26].

The analysis of the evolution of the DESI for the period 2015–2020 at the level of the EU states as a whole shows favorable dynamics for the development of the digital economy and society, and significant heterogeneity in the process of integrating digital technologies [27–29]. In this regard, a number of studies analyze current trends in digital technology, assess the level and trends of the digital economy, and identify homogeneous groups of states based on the development trends of its components among EU countries, concluding that most developed countries have an average level of digitization of the business environment [30].

The digitalization of the economy is a major driver of economic growth and has significant regional business implications [31]. The literature often highlights the impact of the digitization of the economy on the level of GDP, highlighting the dependence of the level of GDP on the level of digitization [32]. The European digital economy is projected to grow 7 times faster than the rest of the economy and will account for 20% of global GDP [33].

A number of studies have focused on determining the main digital factors that can provide the most dynamic economic progress by developing a quantitative model of the impact of digitization on EU economies that can be used to model economic growth [34].

Subsequent research has not addressed the investigation of the impact of e-commerce on the economic development of states (GDP) because the information provided by countries for these aspects is still insufficient.

### 2.2. The Impact of Digitalization on Companies

The digital economy has led to major changes in the way consumers can be involved through innovation and new technologies [35]. Companies use digital technologies to implement and operate new business processes [36,37]. In this sense, IT is considered to be a facilitator of innovation [38]. Digitization and development of digital services have a significant impact on existing business models of companies, influencing to a greater or lesser extent all sectors of activity [36,39]. Practically, the digital economy explores how standard business models are changing [40], analyzing the effects of digitalization on companies' business models, and emphasizing the importance of developing competitive advantages with specific customer service offerings [41]. The digitalization of a company involves the transition from an abstract digital technology to a concrete digital solution, and the digital transformation of companies involves the transition from digital solutions to digital business concepts [37,39]. Due to the use of information and communication technologies, business productivity increases by 5–10% from year to year [42].

The integration of digital technologies at the level of companies produces changes in the operational and organizational environment [43], allowing companies to develop new business opportunities [44], define innovative strategies to address market and new business models [45], and adapt more easily to changing ecosystem conditions [7,46–50].

Although digital transformations can ensure the development of companies [11], they can also be a challenge for SMEs in terms of size and limited resources at their disposal [12]; for this reason, the digitization strategies adopted at the level of SMEs aim at a shorter term compared with that of large companies [13].

### 2.3. E-Commerce

In the digital economy, e-commerce (EC) has become an important technological and distribution tool, stimulating the competitiveness and innovation of companies regarding the way they promote their products and services, producing a transformation in human behavior, and producing a new consumer profile [15]. Through the digital technologies used, e-commerce ensures the direct connection of business partners, increasing the attractiveness of the business environment and new opportunities for consumers [51].

E-commerce is the economic activity of buying and selling products and services through online platforms [52]. Any company can reach global markets with relatively easy-to-use digital e-commerce technology platforms [53].

In the last decade, the use of e-commerce is booming, as most traditional businesses have started to expand their business to the online environment. Thus, a number of studies have approached e-commerce as a basic condition for business development and establishing optimal customer relationships [54], for adopting innovative marketing strategies, new business models, and redefining pricing policy [44,45,55–57].

The adoption of e-commerce by SMEs is influenced by a number of internal, external, and technological factors [58,59]. From the perspective of business globalization, e-commerce allows companies to reduce the distance between them, but also the costs of entering international markets. At the company level, e-commerce optimizes the supply chain, automates internal processes, and provides real-time information on stocks, production, sales, and distribution [60].

At the same time, e-commerce determines financial gains from the perspective of increasing revenues and reducing costs, providing access to a wider range of markets, improved communication, and services for customers [61]. The development of e-commerce may create added value for products and services [62], but the degree of use of EC is still quite low [63].

The development of an e-commerce expansion mechanism for companies involves combining offline and online channels taking into account ways to improve the operational efficiency of business [64]. Developments in e-commerce technologies have enabled companies to improve their customer shopping experiences and their interaction with brands anytime, anywhere [65].

At the level of EU member states, cross-border e-commerce reduces trade costs and generates a positive effect on GDP of EU [66].

### 2.4. The Impact of the COVID-19 Pandemic on E-Commerce

The COVID-19 pandemic has had a significant impact on the EU economy and states. It has significantly changed the role and perception of digitalization in the economies of states and accelerated its pace.

A major consequence of the COVID-19 pandemic on companies has been the acceleration of their digitization programs and the implementation of e-business models, from 35% globally in December 2019 to 55% in July 2020 [67]. The great potential and importance of digital technologies have become even more evident due to their crucial role in the supply chain [68].

The digital technologies used in online commerce have made it possible to overcome the barriers imposed by the COVID-19 pandemic [69] which have accelerated digital transformations in this area [70]. Thus, firms that have intensified the digitization process adapt more easily in the context of the pandemic, whereas digitally immature firms may be vulnerable [71,72].

Existing business models have had to change their strategy, in addition to maintaining traditional methods, by using the digital market, including large e-commerce platforms that have also had to adapt in the context of the COVID-19 crisis. Quarantine, as well as social isolation, were key mental states that led to a significant number of people shopping online. Sales increased by 47% compared with the equivalent period of the year before the pandemic [73]. The number of people who shop online in 2020 has risen to more than 3.4 billion as a result of the pandemic, accounting for more than 43% of consumers worldwide. Projections for 2021 indicate an additional 400 million global e-commerce users by the end of the year [74].

One of the main issues that has supported the exponential growth of online product purchases has been the need to ensure individual safety against the coronavirus [75]. Studies on customer satisfaction with the use of e-commerce during the pandemic show that they are satisfied and will continue to look for this type of trade in the future and in the long run, which will lead to a major technological, social, and economic transformation [76]. Thus, e-commerce continues to grow rapidly, despite global economic uncertainty, with an estimated 22% of total trade in 2023 [77], and e-commerce purchases will account for more than 95% of all purchases by 2040 [74].

The size of the e-commerce market has grown from $9.09 trillion in 2019 to $10.36 trillion in 2020, with global e-commerce projected to have a compound annual growth rate (CAGR) of 14.7% from 2020 to 2027, thus achieving $27.15 trillion in EC revenues in 2027 [78].

The fastest-growing e-commerce in the world is China. With over $1.935 billion in e-commerce sales, China was the largest e-commerce market in the world in 2019. The second-largest e-commerce market in the world is the United States, with $586.9 billion. It is estimated that in 2021 China will become the first country in which more than half of its retail sales will take place online. China is also expected to produce 56.8% of the global total, i.e., over $2.8 trillion from e-commerce sales [79].

At the European level, there is a strong link between the growing popularity of e-commerce and the anxiety of COVID-19 in some countries. The variable number of COVID-19 cases in Europe has been directly reflected in the growing popularity of e-commerce in countries with a high number of cases [80]. Although one year has passed since the lockdown, e-commerce sales in Western Europe will continue to grow significantly by 2025, reaching $758.46 billion (15.9%) of Western European retail sales. The pandemic

has prompted companies to quickly build infrastructure to support increased e-commerce sales. As much as it has grown in 2020, e-commerce is a smaller share in Western Europe than in Asia-Pacific or North America [81].

### 3. Data, Models, and Methodology

The macroeconomic indicators analyzed in this paper are:

- the e-commerce components of the DESI (the percentage of SMEs with online sales—Selling and the share of turnover obtained from EC from the total turnover of SMEs—Turnover);
- the share of gross domestic product from e-commerce in GDP (e-GDP);
- the share of enterprises that are using computer networks for sales of at least 1%—Selling—by types of enterprises (All, Large, SMEs);
- the share of turnover from online sales in the total turnover—Turnover—by types of enterprises (All, Large, SMEs).

The values of these indicators are taken from the Eurostat database [82]. For the DESI indicators, the data are available for the period 2015–2020, whereas for the last two indicators (namely the Selling and Turnover by types of enterprises), the data are available for the period 2003–2020 for most of the EU countries. The values for the e-GDP indicator are extracted from European E-Commerce Report [83].

The study was structured in two parts.

First, for the indicators of DESI and for e-GDP we performed an economic analysis of their evolution over the period 2015–2020. For each country of the EU, we emphasized the trend over the period, the increase in the last year of the period compared with the first one by formula $\frac{x_n - x_1}{x_1} \cdot 100$, and the annual average growth rate (AAGR) given by

$$\text{AAGR} = \frac{1}{n-1} \sum_{k=1}^{n-1} \frac{x_{k+1} - x_k}{x_k} \cdot 100.$$

The AAGR is useful in defining the direction of the trend of growth of the variable. Additionally, it can be used as a measure to forecast the evolution of the variable over the next interval.

Thus, in Section 4, the following economic analyses are performed:

1. analysis of the evolution of EC components of the DESI, namely online Selling and Turnover for small- and medium-sized enterprises in 2015–2020 (Section 4.1);
2. analysis of the evolution of the e-GDP, both as a percentage and in nominal values, over the period 2015–2020 (Section 4.2);
3. comparative analysis of the share of e-commerce enterprises in the total enterprises (Selling) and of the share of turnover in online sales in the total turnover (Turnover) by types of enterprises (2003–2020) (Section 4.3).

The second part of the study aims to perform short-time forecasting for the indicators:

- the e-commerce components of DESI (Selling and Turnover) and e-GDP;
- the share of turnover of EC in total turnover by enterprises (small- and medium-sized, large and all enterprises), using data over the period 2003–2020;
- the percentage of enterprises that are using computer networks for sales of at least 1%, by enterprises (small and medium, large and all enterprises), using data over the period 2003–2020.

For the short-time forecast for 2021 starting with the date over 2015–2020, we used the AAGR. For the data over 2003–2020, the forecast for the horizon 2025 was obtained by using regression models. To this aim, in order to estimate empirically the evolution of each indicator, we used six different regression models. We compared the estimations and chose the best one, according to various statistical indicators. The analysis performed in Section 4.5 relies on the usual statistical instruments [84].

For our estimations, we used three autoregressive models and three polynomial time regression models.

The autoregressive models express the interest variable $y_t$ as a function of previous terms $y_{t-1}, y_{t-2}, \ldots$. As the time series contains a small amount of data, we used only the first- and second-order autoregressive models, namely

$$y_t = a_0 + a_1 y_{t-1} + u_t, \qquad t = 2, \ldots, n, \tag{1}$$

$$y_t = b_0 + b_1 y_{t-1} + b_2 y_{t-2} + u_t, \qquad t = 3, \ldots, n, \tag{2}$$

and the second-order autoregressive model with linear trend

$$y_t = c_0 + c_1 y_{t-1} + c_2 y_{t-2} + c_3 t + u_t, \qquad t = 3, \ldots, n. \tag{3}$$

If $\hat{a}_i$, $\hat{b}_j$, $\hat{c}_k$ are the estimated parameters (using least squares), the predicted values of the variable using the above models satisfy $\hat{y}_t = \hat{a}_0 + \hat{a}_1 \hat{y}_{t-1}$, $\hat{y}_t = \hat{b}_0 + \hat{b}_1 \hat{y}_{t-1} + \hat{b}_2 \hat{y}_{t-1}$, and $\hat{y}_t = \hat{c}_0 + \hat{c}_1 \hat{y}_{t-1} + \hat{c}_2 \hat{y}_{t-1} + \hat{c}_3 t$, respectively.

We also used the linear time regression model

$$y_t = a_0 + a_1 t + u_t, \qquad t = 1, \ldots, n, \tag{4}$$

and the quadratic model

$$y_t = b_0 + b_1 t + b_2 t^2 + u_t, \qquad t = 1, \ldots, n, \tag{5}$$

and the cubic model

$$y_t = c_0 + c_1 t + c_2 t^2 + c_3 t^3 + u_t, \qquad t = 1, \ldots, n, \tag{6}$$

where: $t$ is the discrete time, $y_t$ is the response variable at the time $t$, $a_i$, $b_j$, $c_k$ are the regression coefficients that need to be estimated, $u_t$ is the error term. Although the linear model (4) provides only a general trend for the data, either decreasing, increasing, or stationary, the quadratic and cubic models (5), (6) also give information on the evolution of the variation of the variable.

If $\hat{a}_i$, $\hat{b}_j$, $\hat{c}_k$ are the estimated parameters, the predicted values of the variable are $\hat{y}_t = \hat{a}_0 + \hat{a}_1 t$, or $\hat{y}_t = \hat{b}_0 + \hat{b}_1 t + \hat{b}_2 t^2$, or $\hat{y}_t = \hat{c}_0 + \hat{c}_1 t + \hat{c}_2 t^2 + \hat{c}_3 t^3$, respectively, and the difference $u_t = y_t - \hat{y}_t$, is the residual value at time $t$.

All models are estimated and processed using econometric, data processing, and analysis software EViews 12 [85].

Using the AAGR and the regression models described above, in Section 4 the following short-time forecasts are given:

- short-term forecast by 2021 for DESI (Selling and Turnover) and e-GDP components (Section 4.4);
- forecast by 2025 for Selling (All, Large, and SMEs) and Turnover (All, Large, and SMEs) (Section 4.5).

## 4. Results and Discussion

*4.1. Analysis of the Evolution of EC Components of the DESI (Percentage of SMEs That Sell Online and the Share of Turnover Obtained from EC from the Total Turnover of SMEs)*

Within the DESI, the subcomponents of the e-commerce field have a general upward trend in the period 2015–2020 (Figure 1).

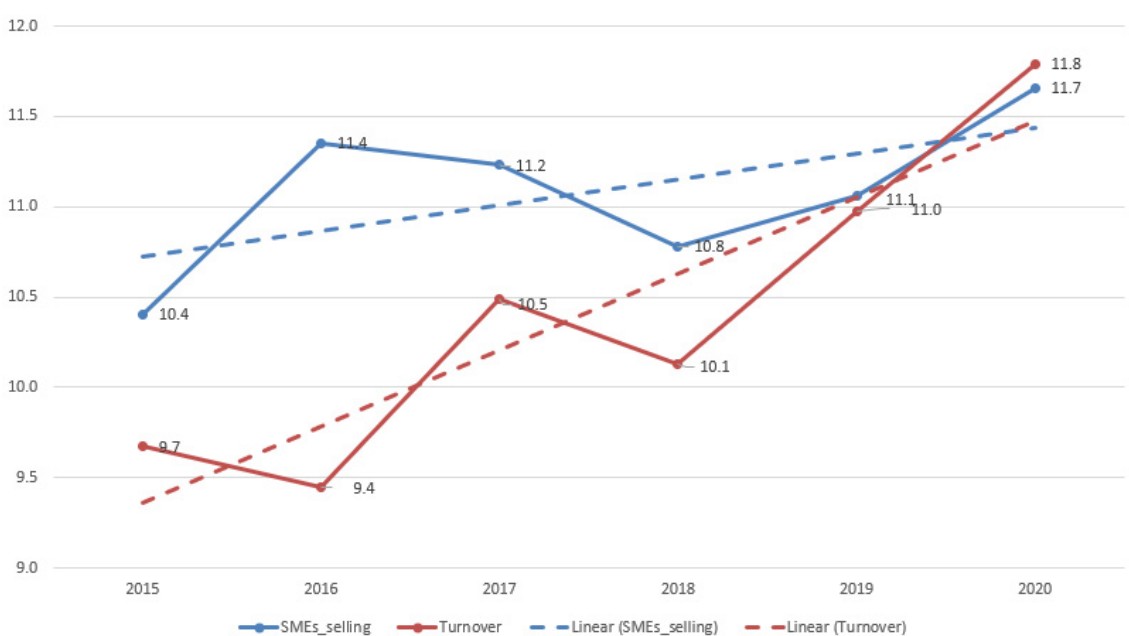

**Figure 1.** Evolution of the Selling and Turnover for EU.

The analysis of the SMEs selling online component (Table A1 in Appendix A) for the period 2015–2020 shows that both in the EU and in most countries the percentage of small- and medium-sized enterprises that sell online has increased. Exceptions are Germany and France, which fell in 2020 compared with 2015.

Germany decreased in 2020 by almost 28% compared with 2015, and France by 18%. The highest increases in the analyzed period were recorded by Romania at 134% in 2020 compared with 2015 with an average annual growth rate of 20.3%, followed by Italy with an increase of 73.2% with an average annual growth rate of 12%. This is followed by a series of six states with increases of over 50% (Croatia, Spain, Lithuania, Malta, Austria, and Estonia) with an average annual growth rate of 9–13%.

At the EU level, the increase was 12%, with an AAGR of 2.4%. Belgium, Ireland, Portugal, and Slovenia recorded increases below the EU average, with the lowest increase being recorded by Portugal at 0.54% and an AAGR of 0.71%.

The use of e-commerce is on the rise in the EU, with the number and share of e-shopping increasing every year. The biggest jump in e-shopping was in 2020 when the COVID-19 pandemic pushed consumers to focus on online shopping. Thus, most EU countries recorded increases in e-commerce turnover share in the total turnover of enterprises in the period 2015–2020 (Table A1 in Appendix A). Exceptions are Bulgaria, Cyprus, Latvia, and Slovakia, which decreased in 2020 compared with 2015, but in the case of Bulgaria and Slovakia, the decreases are very small (less than 1%).

The highest increases were recorded by Greece (248% with an average annual growth rate of 72%), followed by Croatia (96% with an average annual growth rate of 17%), and Malta (95% with an average annual growth rate of 15%).

At the EU level, the increase was 22%, with an AAGR of 4.2%. It can be seen from Table 1 that there are countries that have registered increases below the EU average (Belgium, the Czech Republic, Germany, France, Italy, Portugal, and Sweden), the lowest increase being registered by Belgium with 4.6% and an AAGR of 3%.

**Table 1.** Evolution of the growth rate in 2020 compared with 2015 of the Selling and Turnover (DESI components) on company types.

| Country | Turnover Growth 2020/2015 (%) | | | Selling Growth 2020/2015 (%) | | |
|---|---|---|---|---|---|---|
| | All * | SMEs ** | Large *** | All * | SMEs ** | Large *** |
| AT | 5.872 | 30.182 | −4.595 | 52.921 | 56.592 | 7.851 |
| BE | 23.593 | −2.592 | - | 3.970 | 2.836 | 17.503 |
| BG | 17.080 | −0.922 | 26.064 | 42.159 | 41.404 | 62.702 |
| CY | −6.932 | −29.385 | 121.012 | 39.331 | 41.150 | 2.178 |
| CZ | −0.074 | 7.496 | −3.699 | 25.131 | 26.244 | 11.454 |
| DE | 2.783 | 9.906 | −0.411 | −27.490 | −27.703 | −21.644 |
| DK | 49.568 | 29.553 | 63.300 | 47.479 | 48.967 | 16.358 |
| EE | 7.242 | 51.253 | −24.208 | 34.931 | 34.924 | 46.555 |
| EL | 152.239 | 224.213 | 100.903 | 51.052 | 51.344 | 74.825 |
| ES | 33.510 | 31.281 | 31.812 | 49.643 | 51.143 | 20.843 |
| FI | 8.561 | - | - | 22.561 | 25.098 | −1.898 |
| FR | 31.986 | 12.234 | 37.911 | −17.160 | −18.453 | 0.514 |
| HR | 1.616 | 95.812 | −26.497 | 57.139 | 59.943 | 33.868 |
| HU | 25.241 | 30.050 | 22.860 | 32.381 | 33.330 | 23.269 |
| IE | 17.403 | 40.223 | 14.861 | 1.248 | 2.012 | −9.766 |
| IT | 38.400 | 13.009 | 62.821 | 71.790 | 73.185 | 42.233 |
| LT | 40.915 | 35.181 | 47.144 | 56.683 | 57.189 | 39.037 |
| LU | 8.826 | - | - | 50.535 | 47.176 | 102.801 |
| LV | 43.133 | −13.892 | 259.364 | 37.450 | 35.080 | 62.682 |
| MT | 12.797 | 94.550 | 40.295 | 57.071 | 57.326 | 55.607 |
| NL | 29.785 | 51.608 | 19.242 | 13.082 | 12.756 | 15.561 |
| PL | 28.075 | - | 31.605 | 38.571 | 38.917 | 25.636 |
| PT | 19.784 | 16.937 | 22.126 | 1.313 | 0.536 | 13.407 |
| RO | 60.65796 | 70.60242 | 48.96457 | 130.79 | 134.0434 | 77.72213 |
| SE | 25.78256 | 12.09802 | 32.99721 | 19.632 | 19.46994 | 20.98342 |
| SI | 38.9455 | 38.21625 | 36.62219 | 12.467 | 11.94958 | 27.46201 |
| SK | 3.472638 | −0.34611 | 6.137953 | 32.249 | 33.15176 | 20.89567 |
| EU | 20.70389 | 21.89879 | 19.74521 | 11.675 | 12.07131 | 5.409153 |

Notes: * All—all enterprises (SMEs and Large); ** SMEs—Small and Medium enterprises (10–49 and 50–249 persons employed), without financial sector; *** Large—Large enterprises (250 persons employed or more), without financial sector.

As in previous years, in 2020 Western European countries had the largest share of total European e-commerce turnover.

However, in terms of growth, the countries of Eastern Europe stand out at a rate of 36%, whereas the growth rate of Western Europe remained moderate at 4%.

In recent years, the share of people ordering goods or services online has steadily increased. As a result, businesses have had to supplement their traditional channels with online sales channels, thus, it is found that 20% of EU businesses make online sales, achieving 18% of turnover. The analysis of the share of e-commerce enterprises (Table 1) for the period 2015–2020, shows that in most countries the percentage of enterprises doing e-commerce has increased for all types of enterprises (small, medium, and large). Exceptions are Germany, which saw a significant decline in 2020 compared with 2015, of more than 20% for all business categories, and Finland and Ireland, which saw reductions in online sales at the large enterprises level.

Romania registered the highest increase at the level of all enterprises (131%), followed by Italy at 72%. Ireland and Portugal recorded the smallest increases of around 1%. It can be seen that most states (15) have seen higher increases in the percentage of SMEs that sell electronically compared with large enterprises.

*4.2. Analysis of the Evolution of the Share of GDP from E-Commerce in GDP*

Most EU countries have seen increases in e-commerce turnover as well as in the share of GDP represented by e-commerce (e-GDP). Analyzing the value of GDP from EC

(Figure 2) for the period 2015–2020, it is observed that in most countries the value of e-GDP has increased (except for Germany, which decreased in 2020 compared with 2015).

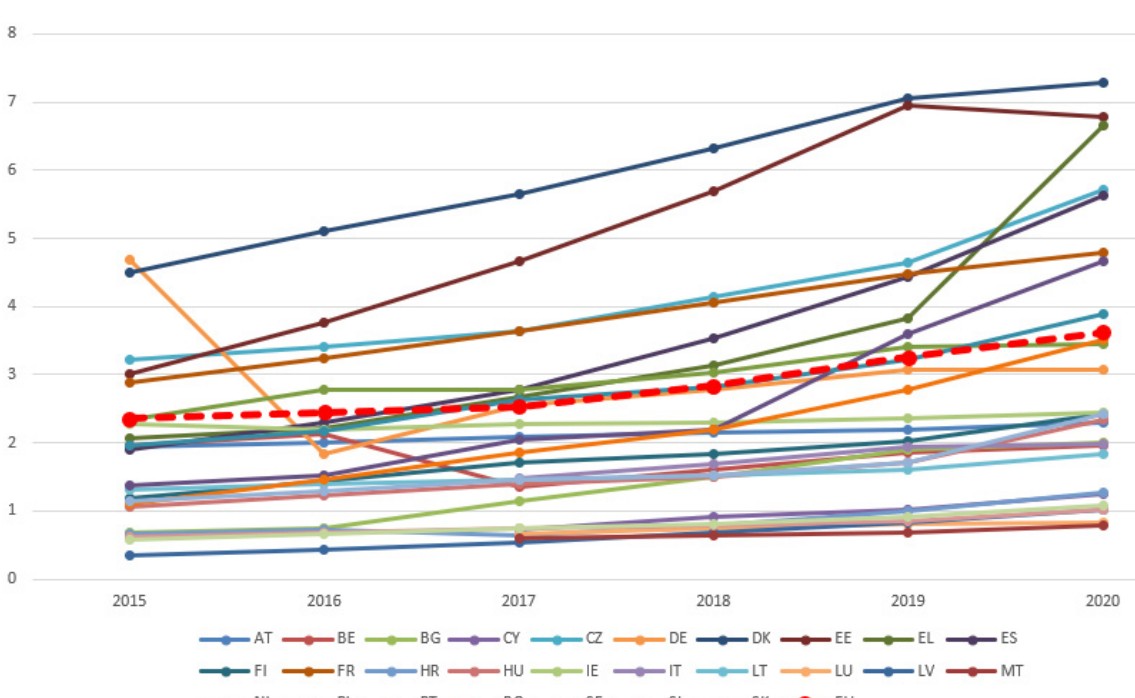

**Figure 2.** Evolution of the e-GDP for period 2015–2020.

Romania and Poland saw the highest increases in e-GDP (over 300% with an average annual growth rate of 34%), and 19 of the EU countries recorded increases of over 50% with an average annual rate between 9% and 33% (Table A2 in Appendix A).

At the EU level, the nominal value of e-GDP increased by 47% with an AAGR of 14%. Austria, Belgium, Italy, Luxembourg, and Malta saw increases below the EU average (the lowest increase was recorded by Belgium with 9% and an AAGR of 4.3%).

The e-GDP analysis as a percentage of total GDP (Table A2 in Appendix A) for the period 2015–2020, shows that in most countries there were increases, except for Belgium and Germany which recorded decreases. Poland and Greece saw the largest increases in the share of e-GDP in GDP (with an increase of over 220%), after which most states (16) recorded increases of over 50% with an average annual growth rate of 10–26%. At the EU level, the increase was 43% with an AAGR of 13%. Growth below the EU average was recorded in Austria, Ireland, Italy, Lithuania, Luxembourg, and Malta (the lowest increase was recorded by Ireland with 7.5% and an AAGR of 1.5%).

*4.3. Comparative Analysis of the Share of E-Commerce Enterprises in the Total Enterprises and of the Share of Turnover from E-Commerce in the Total Turnover by Types of Enterprises (2003–2020)*

EC activity at the EU level in the period 2003–2020 registered an increasing trend for the two components Selling and Turnover. As large companies have greater financial possibilities and have digitized their businesses at a much faster rate than SMEs, the trend for both indicators across all enterprises is determined by the large enterprises (Figure 3).

The analysis of e-commerce activity for the period 2003–2020 at the level of all EU companies (Figure 4) shows that in most countries the percentage of enterprises that sell online has increased. The exception makes Luxembourg, where this percentage fell by around 23% in 2020 compared with 2003.

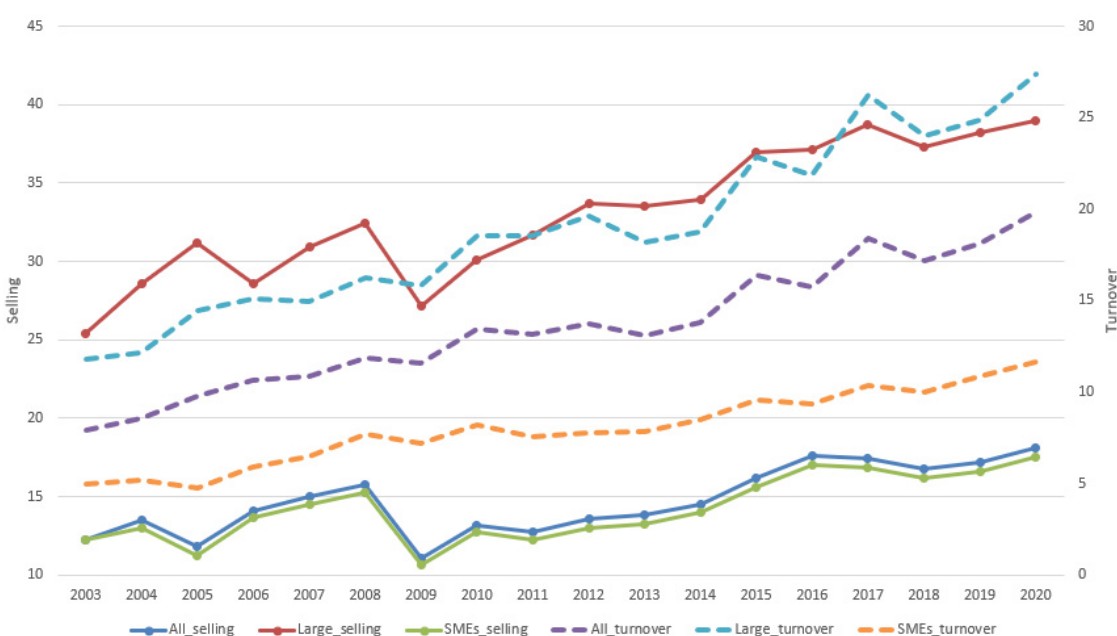

**Figure 3.** Evolution of the indicators for period 2003–2020 for EU.

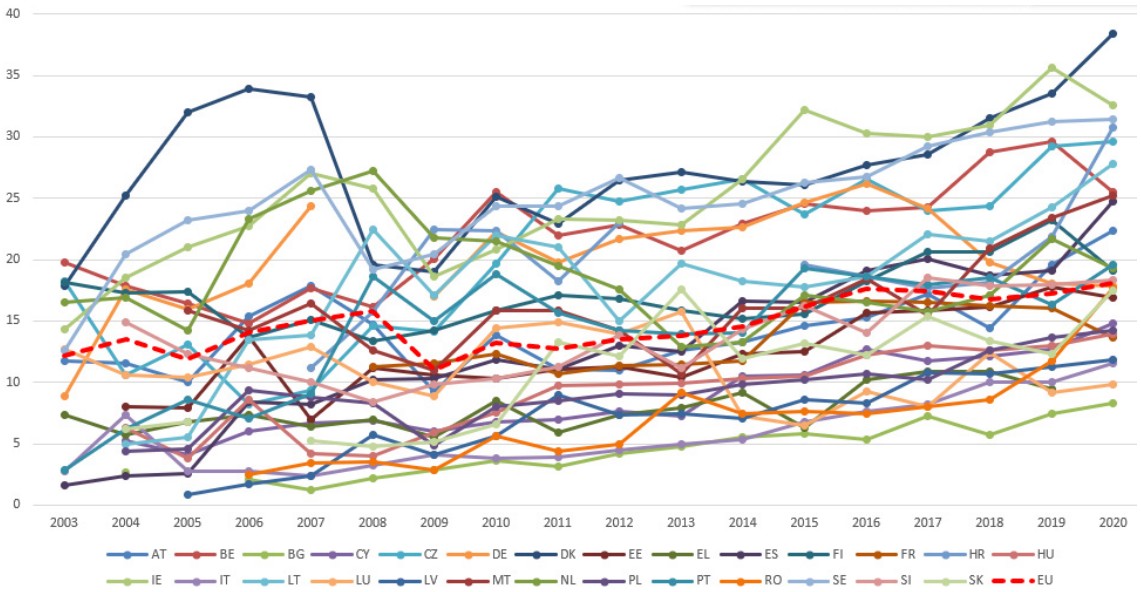

**Figure 4.** Evolution of the percentage of enterprises that use any computer network for sales over the period 2003–2020.

The highest increases were in Spain (over 1400% with an average annual growth rate of 24%), followed by Latvia (over 1200% with an average annual growth rate of 25%). Most EU countries (18) saw increases of over 50% with an average annual growth rate of 3–20% (Table A3 in Appendix A).

At the EU level, the increase in the share of enterprises with EC activity was 34% with an AAGR of 2.6%. Belgium, Greece, Finland, France, the Netherlands, and Slovenia have seen increases below the EU average. Among the EU member states, most companies with EC activity are located in Denmark, Ireland, and Sweden (over 30% of the total number of companies). The share of companies that perform EC and turnover from e-sales varies significantly depending on the size of the enterprises.

At the EU level, in 2020 (Table A4 in Appendix A) the share of large enterprises carrying out EC activity (39%) was more than twice as high as that of SMEs (17.5%).

Sweden and Belgium had the highest shares of large enterprises performing EC (over 60%), whereas the highest shares for SMEs were observed in Denmark and Ireland (over 32%).

The analysis of the share of large enterprises and of SMEs that carry out EC activity (Figures 5 and 6) for the period 2003–2020, shows that in most states they increased. Decreases in 2020 compared with 2003 were recorded in Luxembourg for both categories of enterprises and in Malta for large enterprises.

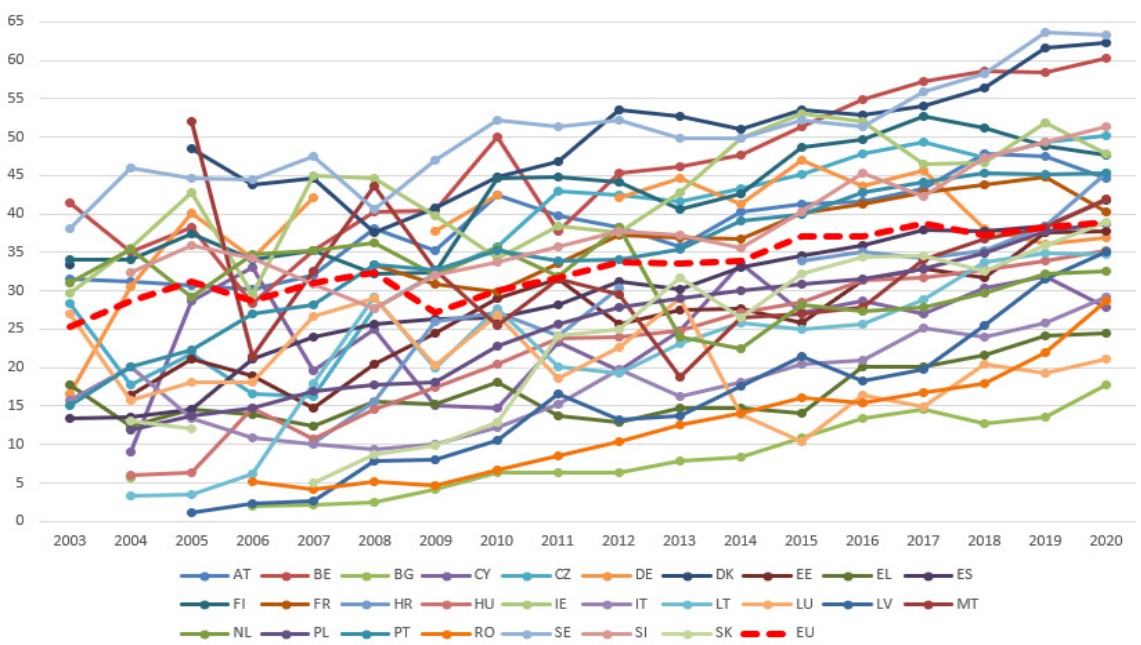

**Figure 5.** Evolution of the percentage of Large enterprises with online sales, compared with EU, over the period 2003–2020.

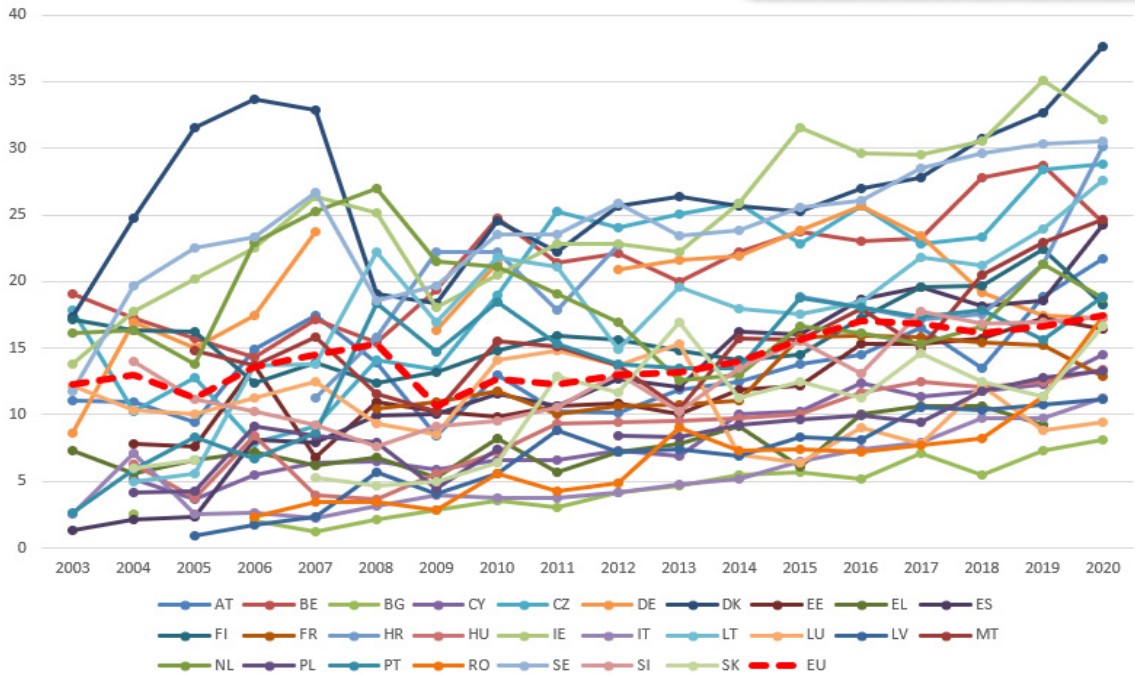

**Figure 6.** Evolution of the percentage of SMEs with online sales, compared with EU, over the period 2003–2020.

Latvia saw the largest increase in the percentage of large enterprises and SMEs selling online (an average annual growth rate of 34% for large enterprises and 25% for SMEs). At the same time, Lithuania saw a significant increase in the percentage of large enterprises carrying out EC (an average annual growth rate of 23%), and for SMEs, Spain saw a significant increase (an average annual growth rate of over 26%).

Most EU countries (17 for large enterprises and 18 for SMEs) saw an increase in the percentage of large enterprises selling online of more than 50%, with average annual growth rates between 3–19% for large enterprises and between 6–20% for SMEs.

In terms of the increase in the percentage of large companies carrying out e-commerce activity, there were increases below the EU average in France and the Netherlands, and for SMEs, Belgium, Greece, Finland, France, the Netherlands, and Slovenia (the smallest increase was recorded by Finland 6.4% and an AAGR of 1%).

The analysis of the turnover component (Tables A3 and A4 in Appendix A) for the period 2003–2020, shows that in most states the percentage of turnover from online sales in the total turnover for all enterprises (small, medium, and large) increased (Figure 7). With the exception of the Czech Republic and Slovakia, for which there were decreases in 2020 compared with 2003.

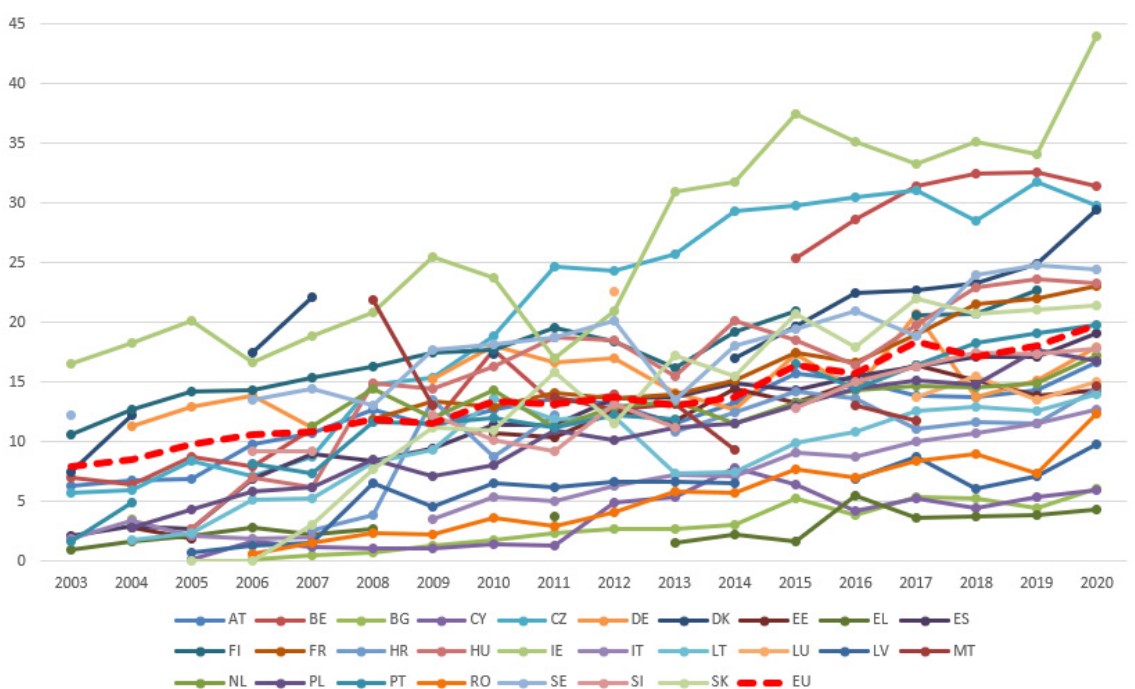

**Figure 7.** Evolution of the Turnover for All enterprises, compared with EU.

For the Turnover component for all enterprises—All—(Table A3 in Appendix A) the highest increase is recorded by Greece at 152%, followed by four states (Romania, Denmark, Lithuania, Latvia) which recorded increases between 40–60%. Nine states have an increase in turnover of over 20%, being above the EU average. The lowest increase, between 1.5% and 3.5%, was recorded by Croatia, Germany, and Slovakia. All the other 23 EU member states that recorded increases exceeded the 50% threshold with an average annual growth rate of 3–42%. Greece maintains its leading position in terms of the highest growth for both SMEs and large enterprises (Figures 8 and 9). For the rest of the states, the increase in turnover varies quite a bit between SMEs and large enterprises. Thus, for SMEs, the highest growth after Greece is recorded by Croatia (although it had the lowest growth of all enterprises) and Malta at over 94%. For most countries, it can be seen that the turnover of large enterprises has increased substantially compared with that of SMEs. Decreases in turnover in 2020 compared with 2003 were recorded for large enterprises in Austria,

the Czech Republic, Germany, Estonia, and Croatia, and for SMEs in Belgium, Bulgaria, Cyprus, Latvia, Poland, and Slovakia.

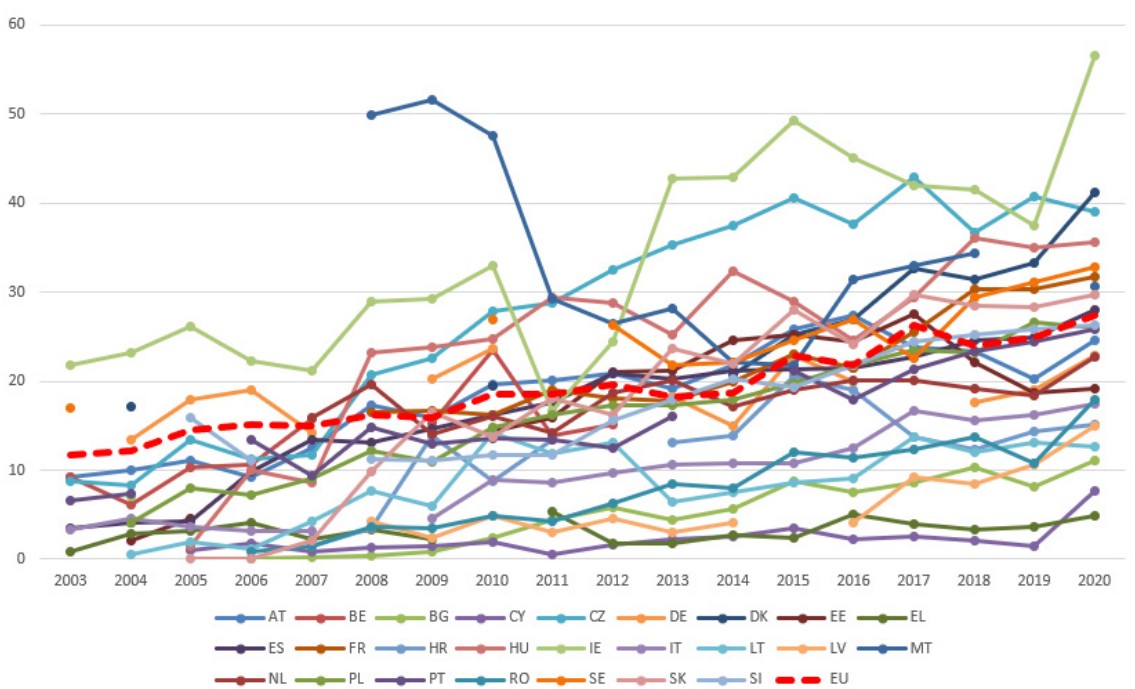

**Figure 8.** Evolution of the Turnover for Large enterprises, compared with EU.

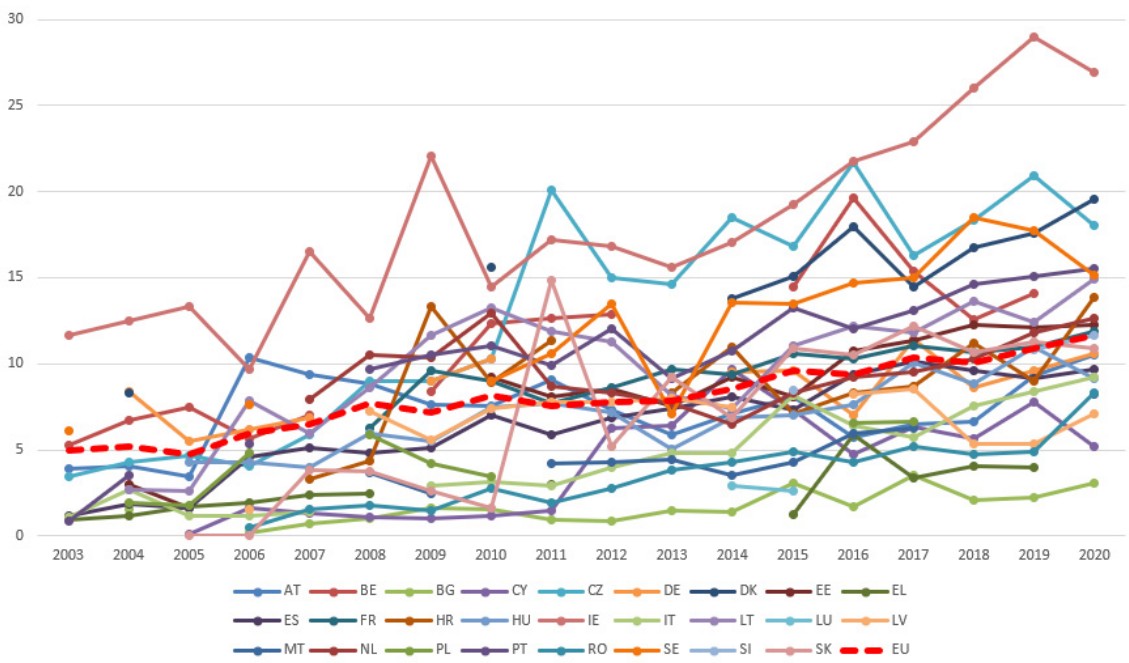

**Figure 9.** Evolution of the Turnover for SMEs, compared with EU.

With regard to the EU, it can be seen that the share of turnover from EC in total is generally around 20% for all categories of companies.

Analyzing the turnover generated by e-sales at the EU level, in 2020 this represented about 20% (19.8%) of the total turnover (Table A5 in Appendix A).

The analysis of this indicator for the period 2003–2020 shows that in most countries the percentage of turnover in online sales in total turnover for all enterprises (small,

medium, and large) increased, with the exception of Luxembourg and Malta which recorded decreases in 2020 compared with 2003.

The largest increase in EC turnover in total turnover was recorded by Slovakia, followed by Cyprus. The other 23 EU member states recorded increases of over 50%, achieving an average annual growth rate of 3–42%. This generated an EU-wide increase of over 131% with an AAGR of 5.7%. Bulgaria, Germany, Finland, France, the Netherlands, Sweden, and Slovenia saw increases below the EU average (the lowest increase was in the Netherlands by 52% and an AAGR of 4.3%).

Among the EU member states, e-commerce-related turnover in 2020 was the highest for businesses in Ireland (over 40% of total turnover), followed by Belgium (over 31%).

The share of e-sales turnover for large enterprises at the EU level in 2020 (Table A5 in Appendix A) was more than twice as high (27%) as for SMEs (11.7%) (Table A4 in Appendix A). The largest companies generating the highest e-sales shares of all member states in 2020 were Ireland (56.5%) and the Czech Republic (40%). The same two member states also reported the largest share of e-sales at SMEs (Ireland 27% and the Czech Republic 18%).

In the period 2003–2020 the percentage of online sales turnover in total turnover for small- and medium-sized and large enterprises, in most EU countries, increased, with the exception of Luxembourg for SMEs and Malta for large enterprises, which decreased in 2020 compared with 2003.

Slovakia recorded the largest percentage increase in EC turnover in total turnover for both categories of enterprises. The other states recorded increases of over 50%.

The increase in the share of EC turnover in total EU turnover for both SMEs and large enterprises was around 124% with an AAGR of 5.6%.

Growth below the EU average at the level share of turnover from EC in total turnover for SMEs was recorded in Bulgaria, Germany, France, Hungary, Malta, the Netherlands, and Slovenia (the lowest increase was recorded by Germany with 26% and AAGR of 3.7%), and for large enterprises in Belgium, Bulgaria, Germany, France, Sweden, and Slovenia (the smallest increase is recorded by Bulgaria with 54%).

### 4.4. Short-Term Forecast for DESI (Selling and Turnover) and e-GDP Components

Using the annual average growth rates (AAGR) determined in Sections 4.1 and 4.2, and considering as a base the values for the year 2020 of the data series, we obtained in Table 2 the empirical estimates for the year 2021 for the values of the DESI components (Selling and Turnover) and e-GDP indicator.

### 4.5. Forecast by 2025 for Selling and Turnover by Type of Enterprises (All, Large, and SMEs)

In this section, we realized an empirical forecast for the macroeconomic indicators presented in Tables A3 and A4 in Appendix A. As the graphs for the period 2003–2020 have obviously different shapes, in order to obtain a good approximation of these shapes we used the regression models (1) to (6). We chose the best estimation by statistical instruments and used this estimation to predict the evolution of each indicator by 2025.

We illustrate the algorithm for the data series of the two indicators Selling and Turnover for all enterprises (All) corresponding to the European Union.

First, we estimated the coefficients of Equations (1)–(6) using as the dependent variable each of the turnover and selling online economic indicators, respectively. The results are reported in Table 3.

**Table 2.** Short-term forecast by 2021 for DESI (Selling and Turnover) and e-GDP components using the average growth rate.

| Cy | Selling (%) | | | Turnover (%) | | | e-GDP(%) | | |
|---|---|---|---|---|---|---|---|---|---|
| | **2020** | **AAGR** | **2021** | **2020** | **AAGR** | **2021** | **2020** | **AAGR** | **2021** |
| AT | 14.45 | 11.01 | 16.04 | 10.58 | 7.92 | 11.42 | 2.30 | 3.47 | 2.38 |
| BE | 16.28 | 1.16 | 16.47 | 15.29 | 3.08 | 15.76 | 1.96 | 2.36 | 2.01 |
| BG | 5.39 | 9.61 | 5.90 | 3.06 | 13.15 | 3.46 | 2.00 | 25.30 | 2.51 |
| CY | 9.68 | 7.64 | 10.42 | 5.24 | −1.50 | 5.16 | 1.24 | 19.44 | 1.48 |
| CZ | 19.18 | 5.35 | 20.21 | 18.22 | 3.42 | 18.85 | 5.71 | 12.37 | 6.42 |
| DE | 11.48 | −5.88 | 10.80 | 10.68 | 6.58 | 11.39 | 3.08 | −0.60 | 3.06 |
| DK | 25.11 | 8.38 | 27.21 | 19.72 | 6.30 | 20.96 | 7.29 | 10.19 | 8.03 |
| EE | 10.93 | 6.68 | 11.66 | 12.36 | 9.25 | 13.51 | 6.78 | 18.13 | 8.01 |
| EL | 6.50 | 12.52 | 7.31 | 4.32 | 71.87 | 7.43 | 6.65 | 28.25 | 8.53 |
| ES | 16.18 | 9.36 | 17.69 | 9.74 | 6.23 | 10.34 | 5.63 | 24.29 | 7.00 |
| FI | 12.14 | 5.53 | 12.81 | 15.63 | 9.16 | 17.06 | 2.43 | 15.69 | 2.81 |
| FR | 8.62 | −3.82 | 8.29 | 12.02 | 2.46 | 12.31 | 4.79 | 10.65 | 5.30 |
| HR | 20.10 | 11.19 | 22.35 | 13.99 | 16.95 | 16.36 | 1.28 | 15.21 | 1.47 |
| HU | 8.92 | 6.11 | 9.46 | 9.21 | 7.14 | 9.87 | 2.33 | 17.23 | 2.73 |
| IE | 21.43 | 0.73 | 21.59 | 27.22 | 7.29 | 29.20 | 2.44 | 1.49 | 2.48 |
| IT | 7.52 | 11.92 | 8.42 | 9.35 | 4.11 | 9.73 | 1.99 | 10.52 | 2.20 |
| LT | 18.42 | 9.76 | 20.21 | 15.07 | 6.79 | 16.09 | 1.84 | 6.94 | 1.97 |
| LU | 6.26 | 12.56 | 7.05 | 3.33 | 5.03 | 3.50 | 0.84 | 7.85 | 0.91 |
| LV | 7.47 | 6.83 | 7.98 | 7.13 | 1.68 | 7.25 | 1.03 | 24.12 | 1.28 |
| MT | 16.42 | 10.85 | 18.20 | 8.34 | 14.86 | 9.58 | 0.79 | 9.11 | 0.86 |
| NL | 12.53 | 3.32 | 12.95 | 12.74 | 8.75 | 13.85 | 3.45 | 10.51 | 3.81 |
| PL | 8.89 | 7.22 | 9.53 | 8.60 | 4.86 | 9.02 | 4.67 | 29.14 | 6.03 |
| PT | 12.61 | 0.71 | 12.70 | 15.68 | 3.44 | 16.22 | 3.90 | 14.78 | 4.48 |
| RO | 11.54 | 20.29 | 13.88 | 8.36 | 14.57 | 9.58 | 3.51 | 25.96 | 4.42 |
| SE | 20.37 | 3.66 | 21.11 | 15.30 | 3.08 | 15.77 | 2.42 | 16.66 | 2.82 |
| SI | 11.55 | 3.59 | 11.97 | 11.80 | 7.31 | 12.66 | 1.01 | 9.99 | 1.11 |
| SK | 11.06 | 8.48 | 12.00 | 10.99 | 0.38 | 11.04 | 1.09 | 13.48 | 1.24 |
| EU | 11.66 | 2.41 | 11.94 | 11.79 | 4.21 | 12.29 | 3.62 | 9.03 | 3.95 |

**Table 3.** Estimations of Equations (1)–(6) for Selling and Turnover for all enterprises for EU data.

| Model | Turnover | | | | | Selling | | | | |
|---|---|---|---|---|---|---|---|---|---|---|
| | **Variable** | **Coeff.** | **Std. Error** | **t-Stat.** | **Prob.** | **Variable** | **Coeff.** | **Std. Error** | **t-Stat.** | **Prob.** |
| Equation (1) | C | 1.3556 | 1.4253 | 0.95 | 0.36 | C | 4.1919 | 3.0899 | 1.36 | 0.20 |
| | All(-1) | 0.9518 | 0.1031 | 9.23 | 0.00 | All(-1) | 0.7334 | 0.2091 | 3.51 | 0.00 |
| Equation (2) | C | 0.9506 | 1.3825 | 0.69 | 0.50 | C | 3.9965 | 3.4641 | 1.15 | 0.27 |
| | All(-1) | 0.3733 | 0.2396 | 1.56 | 0.15 | All(-1) | 0.5745 | 0.2698 | 2.13 | 0.05 |
| | All(-2) | 0.6357 | 0.2353 | 2.70 | 0.02 | All(-2) | 0.1854 | 0.2832 | 0.65 | 0.53 |
| Equation (3) | C | 8.2599 | 0.6569 | 12.57 | 0.00 | C | 12.0763 | 1.1363 | 10.63 | 0.00 |
| | t | 0.6263 | 0.0569 | 11.00 | 0.00 | t | 0.3130 | 0.1470 | 2.13 | 0.05 |
| | AR(1) | −0.1425 | 0.3290 | −0.43 | 0.67 | AR(1) | 0.3033 | 0.4346 | 0.70 | 0.50 |
| | AR(2) | 0.1127 | 0.3475 | 0.32 | 0.75 | AR(2) | 0.0185 | 0.3841 | 0.05 | 0.96 |
| | SIGMASQ | 0.5164 | 0.2568 | 2.01 | 0.07 | SIGMASQ | 1.6710 | 0.8523 | 1.96 | 0.07 |
| Equation (4) | C | 8.2469 | 0.3960 | 20.82 | 0.00 | C | 11.9951 | 0.7365 | 16.29 | 0.00 |
| | t | 0.6278 | 0.0386 | 16.24 | 0.00 | t | 0.3166 | 0.0719 | 4.41 | 0.00 |
| Equation (5) | C | 8.7930 | 0.6382 | 13.78 | 0.00 | C | 13.6538 | 1.1002 | 12.41 | 0.00 |
| | t | 0.4554 | 0.1632 | 2.79 | 0.01 | t | −0.2072 | 0.2814 | −0.74 | 0.47 |
| | $t^2$ | 0.0096 | 0.0088 | 1.09 | 0.30 | $t^2$ | 0.0291 | 0.0152 | 1.92 | 0.08 |
| | C | 7.9392 | 0.9386 | 8.46 | 0.00 | C | 14.1372 | 1.6989 | 8.32 | 0.00 |
| | t | 0.9546 | 0.4387 | 2.18 | 0.05 | t | −0.4899 | 0.7941 | −0.62 | 0.55 |
| | $t^2$ | −0.0578 | 0.0558 | −1.04 | 0.32 | $t^2$ | 0.0673 | 0.1010 | 0.67 | 0.52 |
| | $t^3$ | 0.0025 | 0.0020 | 1.22 | 0.24 | $t^3$ | −0.0014 | 0.0037 | −0.38 | 0.71 |

Note: Prob. is Probability (*p*-value).

In Figure 10 below, we represented the actual values of the two data series for the period 2003–2020 (thicker blue line) and the six regression curves corresponding to the predictions of the estimated models (2021–2025). The simple mean averaging method takes the arithmetic mean of the forecasts at each observation in the forecast sample.

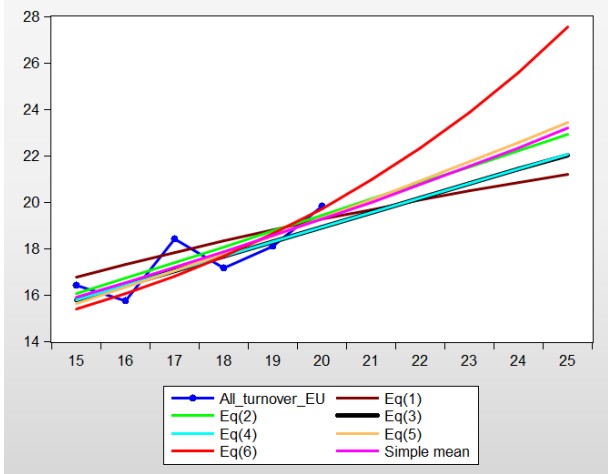 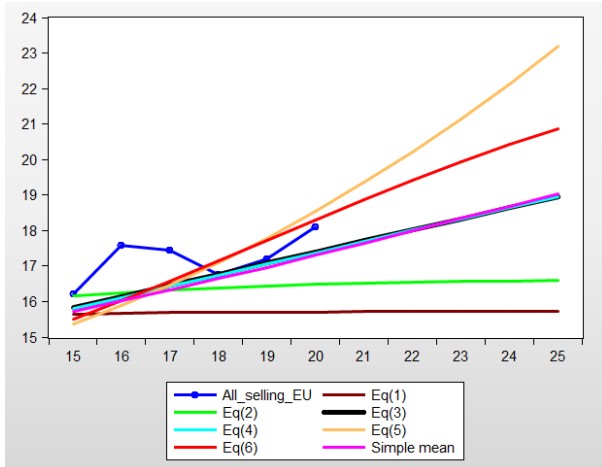

**Figure 10.** Estimation and prediction of total turnover and selling online for EU, horizon 2025, using the autoregressive model of order 1—Equation (1) (brown line), the autoregressive model of order 2—Equation (2) (green line), the ARMA model—Equation (3) (black line), the linear regression—Equation (4) (blue line), the quadratic regression—Equation (5) (yellow line), the cubic regression—Equation (6) (red line).

To compare the quality of the fit of the six models for the analyzed data, we use several regression coefficients, computed by EViews, given in Table 4.

**Table 4.** Fit statistics for models (1)–(6).

| Model | $R^2$ | SSR | AIC | Schwarz Criterion | $R^2$ | SSR | AIC | Schwarz Criterion |
|---|---|---|---|---|---|---|---|---|
| | | | **Turnover** | | | | **Selling** | |
| Equation (1) | 0.859 | 19.714 | 3.297 | 3.393 | 0.468 | 37.541 | 3.941 | 4.037 |
| Equation (2) | 0.897 | 12.209 | 3.032 | 3.174 | 0.486 | 31.040 | 3.965 | 4.107 |
| Equation (3) | 0.948 | 8.779 | 2.768 | 3.013 | 0.608 | 28.408 | 3.946 | 4.191 |
| Equation (4) | 0.946 | 9.141 | 2.453 | 2.551 | 0.564 | 31.613 | 3.694 | 3.792 |
| Equation (5) | 0.950 | 8.430 | 2.489 | 2.636 | 0.655 | 25.049 | 3.578 | 3.725 |
| Equation (6) | 0.956 | 7.560 | 2.498 | 2.694 | 0.658 | 24.770 | 3.685 | 3.881 |

The values of the regression coefficients in Table 4 lead to the conclusion that Equation (6) provides a better fit than the other models. To further compare and choose between estimators to be used for forecasting, we used the forecasting evaluation feature in EViews. This feature offers four different measures of forecast accuracy, namely RMSE (root-mean-square error), MAE (mean absolute error), MAPE (mean absolute percentage error), and the Theil inequality coefficient [85].

For the two tested data series we obtained the following results for the forecast evaluation procedure in Table 5.

These results show that the polynomial cubic regressions Equation (6) statistically provided the best estimation of the data series among the six tested models.

From the data in Table 5, for the cubic estimation Equation (6) of the variable Turnover of the all enterprises (All) of the EU, we find that $\hat{c}_3 > 0$, thus the estimation graph is N-shaped. In addition, there are no turning points, whereas the inflexion time is $-\frac{\hat{c}_2}{3\hat{c}_3} = 7.7$. Consequently, the estimated graph shows a convex increasing trend (see Figure 10, left). For the variable Selling of the all enterprises (All) of the EU, we find that $\hat{c}_3 < 0$, thus the estima-

tion graph has an inverted N-shape. In this case, the turning points are $t_1 = 4.1$, $t_2 = 27.8$, whereas the inflexion time is 16.0, thus the interval of interest for forecasting [19,23] is included in the concave increasing section of the estimation graph (see Figure 10, right). Using these estimations, the expected value for 2025 for the variable Turnover is around 27.54, whereas for the variable Selling it is around 20.45.

**Table 5.** Forecast evaluation for models (1)–(6).

| Model | RMSE | MAE | MAPE | Theil | RMSE | MAE | MAPE | Theil |
|---|---|---|---|---|---|---|---|---|
| | | Turnover | | | | Selling | | |
| Equation (1) | 1.159 | 0.913 | 6.523 | 0.039 | 1.932 | 1.667 | 11.928 | 0.064 |
| Equation (2) | 0.822 | 0.697 | 4.861 | 0.028 | 1.773 | 1.522 | 10.931 | 0.058 |
| Equation (3) | 0.778 | 0.640 | 4.393 | 0.026 | 1.430 | 1.225 | 8.724 | 0.047 |
| Equation (4) | 0.733 | 0.596 | 4.224 | 0.026 | 1.364 | 1.178 | 8.449 | 0.046 |
| Equation (5) | 0.704 | 0.579 | 4.145 | 0.025 | 1.214 | 0.997 | 7.010 | 0.041 |
| Equation (6) | 0.667 | 0.512 | 3.587 | 0.023 | 1.207 | 0.987 | 6.969 | 0.040 |

Applying the same algorithm to the indicators for Selling and Turnover (All, Large, and SMEs) for all the EU countries, and choosing the best statistical estimation for forecasting, we obtained the corresponding expected values for the horizon 2025, displayed in Tables 6 and 7. For most of these indicators, the cubic equation is statistically the best estimate among the six models we tested. The expected values for these have no * in Tables 6 and 7. For the others, the number of * near the value in Tables 6 and 7 is equal to the number of the equation representing the best fit, as explained above.

**Table 6.** Forecast by 2025 for Selling (All, Large, and SMEs).

| Country | All | | | Large | | | SMEs | | |
|---|---|---|---|---|---|---|---|---|---|
| | 2020 | 2025 | Growth | 2020 | 2025 | Growth | 2020 | 2025 | Growth |
| AT | 22.31 | 48.00 | 115.15 | 44.55 | 51.45 | 15.49 | 21.68 | 47.95 | 121.17 |
| BE | 25.54 | 21.74 | −14.88 | 60.31 | 53.47 | −11.34 | 24.42 | 20.37 | −16.58 |
| BG | 8.31 | 10.06 *** | 21.09 | 17.82 | 13.57 | −23.85 | 8.08 | 10.01 ** | 23.89 |
| CY | 14.77 | 18.45 | 24.92 | 27.83 | 33.84 *** | 21.60 | 14.52 | 18.18 | 25.21 |
| CZ | 29.65 | 3.57 | −87.96 | 50.24 | 9.60 | −80.89 | 28.77 | 3.26 | −88.67 |
| DE | 17.86 | 6.08 | −65.96 | 36.88 | 41.33 * | 12.07 | 17.22 | 4.83 | −71.95 |
| DK | 38.40 | 81.94 | 113.39 | 62.23 | 66.58 *** | 6.99 | 37.66 | 81.89 | 117.45 |
| EE | 16.89 | 26.23 | 55.30 | 37.74 | 55.81 | 47.88 | 16.39 | 25.77 | 57.23 |
| EL | - | 12.40 | - | 24.46 | 42.53 | 73.88 | - | 11.43 | - |
| ES | 24.73 | 35.56 | 43.79 | 41.83 | 52.65 | 25.87 | 24.27 | 35.14 | 44.79 |
| FI | 19.12 | 21.52 | 12.55 | 47.71 | 28.34 | −40.60 | 18.21 | 20.77 | 14.06 |
| FR | 13.66 | 16.82 | 23.13 | 40.32 | 8.09 | −79.94 | 12.92 | 15.75 | 21.90 |
| HR | 30.72 | 26.84 | −12.63 | 45.36 | 95.28 | 110.05 | 30.15 | 27.01 | −10.41 |
| HU | 13.96 | 8.33 | −40.33 | 35.20 | 36.93 | 4.91 | 13.37 | 7.53 | −43.68 |
| IE | 32.58 | 53.53 | 64.30 | 47.87 | 47.89 | 0.04 | 32.15 | 53.82 | 67.40 |
| IT | 11.58 | 18.11 | 56.39 | 29.15 | 23.53 | −19.28 | 11.28 | 18.02 | 59.75 |
| LT | 27.83 | 71.69 | 157.60 | 34.65 | 83.09 | 139.80 | 27.62 | 71.29 | 158.11 |
| LU | 9.794 | 10.79 | 10.17 | 21.12 | 42.72 | 102.27 | 9.39 | 9.51 | 1.28 |
| LV | 11.80 | 20.35 | 72.46 | 35.03 | 73.68 | 110.33 | 11.21 | 18.88 | 68.42 |
| MT | 25.25 | 46.79 | 85.31 | 41.89 | 78.30 | 86.92 | 24.63 | 45.45 | 84.53 |
| NL | 19.28 | 61.80 | 220.54 | 32.59 | 68.01 | 108.68 | 18.80 | 61.51 | 227.18 |
| PL | 14.18 | 27.08 | 90.97 | 38.69 | 41.67 | 7.70 | 13.33 | 13.98 *** | 4.88 |
| PT | 19.56 | 28.06 | 43.46 | 45.26 | 61.39 | 35.64 | 18.92 | 27.20 | 43.76 |
| RO | 17.67 | 42.03 | 137.86 | 28.71 | 47.41 | 65.13 | 17.30 | 41.96 | 142.54 |
| SE | 31.45 | 34.44 *** | 9.51 | 63.21 | 93.03 | 47.18 | 30.55 | 33.63 *** | 10.08 |
| SI | 18.30 | 9.09 | −50.33 | 51.36 | 60.91 | 18.59 | 17.33 | 7.91 | −54.36 |
| SK | 17.46 | 2.13 | −87.80 | 38.87 | 41.62 | 7.07 | 16.60 | 2.51 | −84.88 |
| EU | 18.10 | 20.87 | 15.30 | 39.00 | 35.35 | −9.36 | 17.49 | 20.45 | 16.92 |

Notes: * Equation (1), ** Equation (2), *** Equation (3).

**Table 7.** Forecast by 2025 for Turnover (All, Large, and SMEs).

| Country | All | | | Large | | | SMEs | | |
|---|---|---|---|---|---|---|---|---|---|
| | **2020** | **2025** | **Growth** | **2020** | **2025** | **Growth** | **2020** | **2025** | **Growth** |
| AT | 16.72 | 21.13 | 26.38 | 24.57 | 7.01 | −71.47 | 10.48 | 30.08 | 187.02 |
| BE | 31.41 | 29.75 | −5.28 | - | 14.34 | - | - | 13.15 | - |
| BG | 6.12 | 6.50 ** | 6.21 | 11.01 | 14.34 ** | 30.25 | 3.03 | 1.46 | −51.82 |
| CY | 5.95 | 5.18 | −12.94 | 7.65 | 15.15 | 98.04 | 5.19 | 5.31 | 2.31 |
| CZ | 29.80 | 3.73 | −87.48 | 39.00 | 7.37 | −81.10 | 18.04 | 2.88 | −84.04 |
| DE | 17.90 | 22.54 | 25.92 | 22.80 | 37.28 | 63.51 | 10.58 | 11.13 | 5.20 |
| DK | 29.47 | 67.24 | 128.16 | 41.21 | 69.62 | 68.94 | 19.52 | 37.12 | 90.16 |
| EE | 14.29 | 4.36 | −69.49 | 19.09 | 22.75 | 19.17 | 12.24 | 19.81 | 61.85 |
| EL | 4.31 | 9.72 | 125.52 | 4.94 | 11.32 | 129.15 | - | 7.83 | - |
| ES | 19.14 | 22.99 | 20.11 | 28.04 | 35.79 | 27.64 | 9.64 | 10.85 | 12.55 |
| FR | 23.07 | 33.14 | 43.65 | 31.72 | 45.74 | 44.20 | 13.85 | 13.16 *** | −4.98 |
| HR | 14.42 | 12.40 ** | −14.01 | 15.06 | 14.76 | −1.99 | 9.12 | 53.85 | 490.46 |
| HU | 23.25 | 47.96 | 106.28 | 35.57 | 76.01 | 113.69 | 26.95 | 16.02 | −40.56 |
| IE | 43.95 | 44.79 | 1.91 | 56.51 | 43.90 | −22.31 | 9.26 | 48.94 | 428.51 |
| IT | 12.68 | 13.77 | 8.60 | 17.43 | 18.22 | 4.53 | 14.92 | 10.13 | −32.10 |
| LT | 13.95 | 31.80 | 127.96 | 12.57 | 27.02 | 114.96 | - | 34.89 | - |
| LV | 9.82 | 18.83 | 91.75 | 30.67 | 43.58 | 42.09 | 8.26 | 9.10 | 10.17 |
| MT | 14.71 | 11.63 | −20.94 | 22.67 | 29.74 ** | 31.19 | 12.61 | 14.68 | 16.42 |
| NL | 17.29 | 29.47 | 70.45 | 26.04 | 22.83 *** | −12.33 | - | 39.33 | - |
| PL | 16.79 | 23.62 | 40.68 | 25.83 | 32.56 ** | 26.05 | 15.53 | 7.81 *** | −49.71 |
| PT | 19.85 | 34.80 | 75.31 | 17.85 | 43.77 | 145.21 | 8.28 | 28.73 | 246.98 |
| RO | 12.42 | 17.44 | 40.42 | 32.79 | 21.32 | −34.98 | 15.15 | 14.01 | −7.52 |
| SE | 24.44 | 37.64 | 54.01 | 26.32 | 74.06 | 181.38 | 11.68 | 16.35 | 39.98 |
| SI | 17.81 | 22.51 | 26.39 | 29.66 | 11.57 | −60.99 | 10.88 | 18.07 ***** | 66.08 |
| SK | 21.43 | 23.11 ** | 7.84 | 27.36 | 26.29 | −3.91 | 11.67 | 5.81 | −50.21 |
| EU | 19.83 | 27.54 | 38.88 | 24.57 | 36.16 | 47.17 | 10.48 | 17.44 | 66.41 |

Notes: ** Equation (2) *** Equation (3), ***** Equation (5).

For Selling (Table 6), the e-commerce market has matured for the Large enterprises, as a result of which there is a substantially lower estimated growth than that of SMEs in 2025, compared with 2020. At the EU level, there is a decrease of over 9% in Large and an increase of around 17% in SMEs. The main increases estimated according to the forecast models used are registered with over 100% in Lithuania, Latvia, Croatia, Luxembourg, and the Netherlands for Large, respectively, and the Netherlands, Lithuania, Romania, Austria, and Denmark for SMEs. In addition, the lowest estimated increases are observed in the case of Large in Ireland, Hungary, Denmark, Slovakia, and Poland (0.04–7.70%), respectively, and for SMEs in Luxembourg (1.28%) and Poland (4.88%).

For the Turnover indicator (Table 7), the models used empirically estimate an increase in 2025 compared with 2020, at the EU level, for Large (47%), and especially for SMEs (66%). Sweden, Portugal, Greece, Lithuania, and Hungary are expected to increase by more than 100% for Large, respectively, whereas Croatia, Ireland, Portugal, and Austria even more (187–490%) for SMEs, respectively. The lowest increases are found in the case of Large in Italy (5%), and for SMEs in Cyprus, Germany, and Latvia (2–10%).

## 5. Conclusions

Creating a Europe-friendly digital age is a top priority for the European Commission. EU e-commerce rules break down barriers so that people may enjoy full access to all goods and services offered online by EU businesses.

E-commerce has a profound impact and is changing the business environment in general, and especially the way business is conducted around the world. The motivating factor for this study was the lack of research on the degree of adoption of e-commerce by businesses and its impact on their turnover and society in general.

Therefore, the main purpose of this research was to examine the evolution and provide a forecast for the percentage of e-commerce enterprises and their turnover, as the main sub-

components of the integration of digital technologies by enterprises (the third component of the DESI), as well as the impact on the development of national economies by increasing the share of GDP obtained from e-commerce in total GDP.

In this sense, the study was oriented in two directions. The first direction aimed at conducting economic analyses on: (1) the evolution in the period 2015–2020 of the components of the DESI (global and by enterprises); (2) the evolution of e-GDP in GDP (nominal and as a percentage) for the period 2015–2020; (3) comparative analysis of the share of e-commerce enterprises in total enterprises and the share of turnover in online sales in total turnover by type of enterprise for the period 2003–2020. The second direction of the study focused on providing forecasts to determine the evolution trend of the variables: (1) two components of the DESI and the e-GDP only for 2021 (due to lack of data); (2) Selling and Turnover indicators by types of enterprises for 2025.

The analysis of the evolution of the variables in the DESI (SMEs, Selling, and Turnover) for the period 2015–2020 both at the EU level and in most states shows a favorable dynamic for the development of e-commerce, but there is significant heterogeneity in the integration of e-commerce by countries and by type of enterprise, which confirms the results of the other research [27–29].

The growth rate of the Selling variable shows that e-commerce has already matured in developed countries (Western Europe), which have lower growth rates (even negative in the case of Germany), and is still growing in other countries (Eastern Europe), which have higher growth rates but started at a lower level.

The results of the analysis show that the share of companies performing e-commerce and the turnover in e-sales vary significantly depending on the size of the enterprises. It is important to note that in most countries there is a more significant increase in the percentage of SMEs that make electronic sales compared to large enterprises, but for most countries, it is observed that the turnover of large enterprises has increased substantially compared with that of SMEs. During the period under review, it is noted that Western European countries have the highest share of total European e-commerce turnover, but have a moderate growth rate compared with Eastern European countries, which, although they have a significantly higher growth rate, the total turnover of European e-commerce is lower, results which are confirmed by another research [30].

The results of this study are in line with previous research [32,33], demonstrating the impact of e-commerce use by businesses on increasing the level of GDP by constantly increasing the share of GDP that comes from electronic transactions in all states. The maturity of the e-commerce market in developed countries is also noticeable here, which, although they have a slow growth rate of e-GDP, the nominal value is significant compared with the less developed countries.

The results of the 2021 forecast are in accordance with the findings [77], which confirmed that e-commerce will continue to grow at a rapid pace, despite global economic uncertainty. But it should not be overlooked that growth in 2021 is likely to be significantly higher, especially in mature markets, which have had the ability to expand their e-commerce, coupled with increased demand during the coronavirus pandemic.

Following the forecast for the Selling and Turnover macroeconomic indicators by types of enterprises, using six regression models, we found that the best estimate for most states is the time cubic polynomial estimator.

The COVID-19 pandemic has brought a lot of rapid changes, one of which has been on how to do business at the enterprise level, causing them to adapt quickly to the new way of trading—e-commerce. According to the results of the forecast, e-commerce is not only thriving but is expected to grow significantly in most EU countries by 2025. The results of the study support previous studies [18,21,44,45,56,57], according to which the majority of traditional businesses have begun to expand their business to the online environment, a growth that will continue in the coming years, according to forecasts.

E-commerce can be a major means by which SMEs can compete with their larger counterparts with the potential to expand into global markets. The current situation of the COVID-19 pandemic has accelerated e-commerce and e-commerce popularity worldwide.

The results offer important implications both for enterprises (regardless of their size, but especially for SMEs) and for society in general, by the increase of the share of e-commerce turnover in the total turnover of enterprises and implicitly on e-GDP. The results show that companies, regardless of their size, are in a continuous process of adopting a new form of sales—e-commerce.

From this study, we can derive some recommendations for both companies and governments of EU member states.

First, companies need to accelerate the pace of digital transformation and e-commerce adoption to proactively bring about change. At the same time, it is necessary for companies to be aware that e-commerce is an important technological and distribution tool, which stimulates competitiveness and innovation to promote their products and services, and they can easily reach global markets. The crisis triggered by the COVID-19 pandemic requires companies to accelerate the processes of digitization and implementation of e-business models to ensure the resilience of the business in conditions of uncertainty.

At the governmental level, EU countries need to pay more attention to supporting businesses to increase their share of turnover obtained from e-commerce in total turnover, which contributes substantially to increasing national income.

At the same time, this study can contribute to theoretical research and can contribute to filling in the gap in the literature on the analysis of companies that carry out e-commerce activity. Business activity has been severely affected by the COVID-19 pandemic. The pre-COVID and post-COVID situations are completely different and adapting business to a new style of work is difficult. But their adaptation to e-commerce is the test that every company should pass with flying colors.

The current preliminary results will be used in the design of a future in-depth analysis of e-commerce trends and the new post-COVID-19 pandemic reality and the impact on the development of each economy. There are some limitations of the current study, on the one hand on the time horizon for which the data on e-GDP and components of the DESI were available, and on the other hand the lack of information on the value of e-commerce transactions by type of enterprise, which could have allowed us to determine the impact of adopting e-commerce by enterprises on the development of a country's economy.

**Author Contributions:** Conceptualization, G.S. and A.M.; formal analysis, G.S. and A.M.; investigation, G.S. and A.M.; methodology, G.S., A.M. and M.S.; resources, G.S. and A.M.; software, G.S. and A.M.; validation, G.S., A.M. and M.S.; writing—original draft, G.S., A.M. and M.S. All authors have read and agreed to the published version of the manuscript.

**Funding:** This research received no external funding.

**Institutional Review Board Statement:** Not applicable.

**Informed Consent Statement:** Not applicable.

**Data Availability Statement:** The series for the indicators was extracted from Eurostat: https://digital-agenda-data.eu/datasets/desi/indicators (accessed on 14 November 2021) [82] and from European E-Commerce Report: https://ecommerce-europe.eu/wp-content/uploads/2021/09/2021-European-E-commerce-Report-LIGHT-VERSION.pdf (accessed on 6 November 2021) [83].

**Acknowledgments:** The authors acknowledge the anonymous reviewers whose suggestions and comments helped improve the paper.

**Conflicts of Interest:** The authors declare no conflict of interest.

## Appendix A

**Table A1.** Trend and growth rate analysis for SMEs Selling and Turnover (DESI components).

| Cy | Selling | | | | Turnover | | | |
|---|---|---|---|---|---|---|---|---|
| | Trend 2015–2020 | 2020/2015 (%) | | AAGR 2015–2020 | Trend 2015–2020 | 2020/2015 (%) | | AAGR 2015–2020 |
| AT | | 56.59 | 11.01 | | | 30.18 | 7.92 | |
| BE | | 2.84 | 1.16 | | | 4.60 | 3.08 | |
| BG | | 41.40 | 9.61 | | | −0.92 | 13.15 | |
| CY | | 41.15 | 7.64 | | | −29.38 | −1.50 | |
| CZ | | 26.24 | 5.35 | | | 7.50 | 3.42 | |
| DE | | −27.70 | −5.88 | | | 9.91 | 6.58 | |
| DK | | 48.97 | 8.38 | | | 29.55 | 6.30 | |
| EE | | 34.92 | 6.68 | | | 51.25 | 9.25 | |
| EL | | 59.54 | 12.52 | | | 248.14 | 71.87 | |
| ES | | 51.14 | 9.36 | | | 31.28 | 6.23 | |
| FI | | 25.10 | 5.53 | | | 51.32 | 9.16 | |
| FR | | −18.45 | −3.82 | | | 12.23 | 2.46 | |
| HR | | 59.94 | 11.19 | | | 95.81 | 16.95 | |
| HU | | 33.33 | 6.11 | | | 30.05 | 7.14 | |
| IE | | 2.01 | 0.73 | | | 40.22 | 7.29 | |
| IT | | 73.19 | 11.92 | | | 13.01 | 4.11 | |
| LT | | 57.19 | 9.76 | | | 35.18 | 6.79 | |
| LU | | 47.18 | 12.56 | | | 27.16 | 5.03 | |
| LV | | 35.08 | 6.83 | | | −5.75 | 1.68 | |
| MT | | 57.33 | 10.85 | | | 94.55 | 14.86 | |
| NL | | 12.76 | 3.32 | | | 51.61 | 8.75 | |
| PL | | 38.92 | 7.22 | | | 26.06 | 4.86 | |
| PT | | 0.54 | 0.71 | | | 16.94 | 3.44 | |
| RO | | 134.04 | 20.29 | | | 70.60 | 14.57 | |
| SE | | 19.47 | 3.66 | | | 12.10 | 3.08 | |
| SI | | 11.95 | 3.59 | | | 38.22 | 7.31 | |
| SK | | 33.15 | 8.48 | | | −0.35 | 0.38 | |
| EU | | 12.07 | 2.41 | | | 21.90 | 4.21 | |

Notes: Cy—country; AT—Austria, BE—Belgium, BG—Bulgaria, CY—Cyprus, CZ—the Czech Republic, DE—Germany, DK—Denmark, EE—Estonia, EL—Greece, ES—Spain, FI—Finland, FR—France, HR—Croatia, HU—Hungary, IE—Ireland, IT—Italy, LT—Lithuania, LU—Luxembourg, LV—Latvia, MT—Malta, NL—Netherlands, PL—Poland, PT—Portugal, RO—Romania, SE—Sweden, SI—Slovenia, SK—Slovakia, EU—European Union.

**Table A2.** Trend and average annual growth rate for e-GDP.

| Cy | e-GDP(Billion Euro) | | | e-GDP(% from GDP) | | |
|----|---|---|---|---|---|---|
| | Trend 2015–2020 | 2020/2015 (%) | AAGR 2015–2020 | Trend 2015–2020 | 2020/2015 (%) | AAGR 2015–2020 |
| AT | | 30.63 | 5.53 | | 18.56 | 3.47 |
| BE | | 9.09 | 4.26 | | −0.51 | 2.36 |
| BG | | 293.75 | 33.25 | | 194.12 | 25.30 |
| CY | | 80.80 | 22.09 | | 69.86 | 19.44 |
| CZ | | 125.82 | 17.77 | | 77.88 | 12.37 |
| DE | | −26.76 | 1.73 | | −34.19 | −0.60 |
| DK | | 85.44 | 13.25 | | 62.00 | 10.19 |
| EE | | 192.97 | 25.06 | | 125.25 | 18.13 |
| EL | | 202.60 | 25.81 | | 222.82 | 28.25 |
| ES | | 208.51 | 25.42 | | 196.32 | 24.29 |
| FI | | 130.08 | 18.27 | | 105.93 | 15.69 |
| FR | | 73.62 | 11.80 | | 65.74 | 10.65 |
| HR | | 115.41 | 17.45 | | 93.94 | 15.21 |
| HU | | 163.69 | 21.47 | | 117.76 | 17.23 |
| IE | | 52.51 | 8.93 | | 7.49 | 1.49 |
| IT | | 28.03 | 9.10 | | 34.46 | 10.52 |
| LT | | 84.79 | 13.08 | | 39.39 | 6.94 |
| LU | | 38.42 | 11.49 | | 25.37 | 7.85 |
| LV | | 253.44 | 28.88 | | 194.29 | 24.12 |
| MT | | 41.37 | 12.28 | | 29.51 | 9.11 |
| NL | | 71.69 | 11.75 | | 48.07 | 10.51 |
| PL | | 311.68 | 34.78 | | 238.41 | 29.14 |
| PT | | 120.41 | 17.25 | | 97.97 | 14.78 |
| RO | | 330.77 | 34.10 | | 216.22 | 25.96 |
| SE | | 119.58 | 17.66 | | 110.43 | 16.66 |
| SI | | 93.60 | 14.13 | | 60.32 | 9.99 |
| SK | | 116.61 | 16.72 | | 87.93 | 13.48 |
| EU | | 46.56 | 13.72 | | 43.08 | 12.69 |

**Table A3.** Trend, growth rate, and average annual growth rate for percentage of all enterprises which use any computer network for sales and percentage electronic commerce turnover of their total turnover.

| Cy | Selling (All) | | | Turnover (All) | | |
|---|---|---|---|---|---|---|
| | Trend 2003–2020 | 2020/ 2003 (%) | AAGR (%) 2003–2020 | Trend 2003–2020 | 2020/ 2003(%) | AAGR (%) 2003–2020 |
| AT | | 90.17 | 6.46 | | 165.86 | 6.69 |
| BE | | 29.53 | 2.36 | | 346.81 | 9.64 |
| BG | | 208.86 | 11.51 | | 70.55 | 41.66 |
| CY | | 181.41 | 8.00 | | 2731.05 | 66.06 |
| CZ | | 62.91 | 5.86 | | 420.97 | 11.78 |
| DE | | 102.01 | 3.14 | | 57.70 | 3.28 |
| DK | | 115.13 | 6.25 | | 291.92 | 18.51 |
| EE | | 111.66 | 8.18 | | 412.29 | 0.42 |
| EL | | 28.34 | −1.40 | | 363.39 | 30.55 |
| ES | | 1445.86 | 24.10 | | 802.68 | 17.39 |
| FI | | 5.61 | 0.91 | | 114.12 | 6.08 |
| FR | | 21.78 | 2.41 | | 93.09 | 5.87 |
| HR | | 174.31 | 14.43 | | 444.24 | 30.23 |
| HU | | 121.28 | 10.67 | | 751.67 | 22.76 |
| IE | | 127.22 | 5.84 | | 164.61 | 7.37 |
| IT | | 316.55 | 15.60 | | 556.80 | 12.29 |
| LT | | 464.57 | 16.30 | | 670.70 | 17.97 |
| LU | | −22.64 | 2.55 | | −33.51 | 3.52 |
| LV | | 1256.33 | 25.40 | | 1283.54 | 37.72 |
| MT | | 59.32 | 5.46 | | −32.73 | −16.50 |
| NL | | 16.42 | 2.76 | | 51.96 | 4.28 |
| PL | | 221.61 | 11.22 | | 497.35 | 13.33 |
| PT | | 574.48 | 16.97 | | 1110.27 | 20.89 |
| RO | | 612.51 | 19.74 | | 2004.86 | 31.03 |
| SE | | 150.39 | 6.86 | | 98.73 | 5.78 |
| SI | | 23.16 | 2.56 | | 93.76 | 3.64 |
| SK | | 180.32 | 13.63 | | 107033.50 | 2095.81 |
| EU | | 34.34 | 2.56 | | 131.39 | 5.68 |

**Table A4.** Trend, growth rate, and average annual growth rate for percentage of enterprises (SMEs and Large) which use any computer network for sales.

| Cy | Selling (SMEs) | | | Selling (Large) | | |
|---|---|---|---|---|---|---|
| | Trend 2003–2020 | Growth 2020/2003(%) | AAGR 2003–2020 | Trend 2003–2020 | Growth 2020/2003(%) | AAGR 2003–2020 |
| AT | | 95.81 | 7.11 | | 41.30 | 2.39 |
| BE | | 28.07 | 2.34 | | 45.39 | 3.25 |
| BG | | 213.17 | 13.98 | | 209.86 | 18.99 |
| CY | | 180.37 | 8.35 | | 206.21 | 16.56 |
| CZ | | 61.28 | 5.92 | | 77.02 | 5.47 |
| DE | | 99.73 | 11.74 | | 122.57 | 9.71 |
| DK | | 117.21 | 6.42 | | 86.14 | 1.97 |
| EE | | 110.19 | 8.39 | | 130.38 | 6.64 |
| EL | | 26.70 | −1.32 | | 37.17 | 3.35 |
| ES | | 1684.32 | 26.25 | | 214.07 | 7.38 |
| FI | | 6.39 | 1.04 | | 39.78 | 2.49 |
| FR | | 23.91 | 2.66 | | 20.75 | 1.80 |
| HR | | 168.24 | 14.50 | | 347.76 | 17.09 |
| HU | | 111.61 | 10.61 | | 480.82 | 14.95 |
| IE | | 133.66 | 6.04 | | 60.92 | 4.33 |
| IT | | 337.16 | 16.57 | | 84.71 | 5.27 |
| LT | | 455.83 | 16.25 | | 966.26 | 23.48 |
| LU | | −23.16 | 2.63 | | −21.56 | 3.70 |
| LV | | 1188.14 | 25.03 | | 3204.84 | 34.19 |
| MT | | 66.40 | 5.97 | | −19.43 | 3.40 |
| NL | | 16.24 | 2.84 | | 4.75 | 1.51 |
| PL | | 221.93 | 12.63 | | 225.66 | 7.84 |
| PT | | 603.26 | 17.74 | | 200.50 | 7.06 |
| RO | | 630.16 | 20.40 | | 455.29 | 14.27 |
| SE | | 160.44 | 7.24 | | 65.91 | 3.33 |
| SI | | 24.13 | 2.86 | | 58.46 | 3.26 |
| SK | | 176.62 | 13.66 | | 197.18 | 18.17 |
| EU | | 34.74 | 2.64 | | 36.36 | 2.19 |

**Table A5.** Trend, growth rate, and average annual growth rate for turnover from the EC by enterprises (SMEs and Large), as a percentage of their total turnover.

| Cy | Turnover (SMEs) | | | Turnover (Large) | | |
|----|------|------|------|------|------|------|
| | Trend 2003–2020 | 2020/ 2003 (%) | AAGR 2003–2020 (%) | Trend 2003–2020 | 2020/ 2003 (%) | AAGR 2003–2020 (%) |
| AT | | 167.32 | 12.53 | | 168.52 | 7.17 |
| BE | | 166.49 | −0.11 | | 65.67 | 15.40 |
| BG | | 96.85 | 42.47 | | 54.47 | 70.44 |
| CY | | 3606.86 | 92.07 | | 722.14 | 45.84 |
| CZ | | 429.04 | 13.87 | | 347.74 | 11.34 |
| DE | | 25.63 | 3.73 | | 70.81 | 9.19 |
| DK | | 134.87 | 6.82 | | 141.44 | 12.61 |
| EE | | 313.51 | −0.59 | | 813.56 | 14.35 |
| EL | | 328.62 | 36.31 | | 437.15 | 21.15 |
| ES | | 710.06 | 18.81 | | 719.75 | 15.94 |
| FR | | 89.14 | 6.54 | | 92.84 | 6.03 |
| HR | | 323.49 | 28.66 | | 335.32 | 42.82 |
| HU | | 114.51 | 7.48 | | 314.53 | 53.63 |
| IE | | 131.69 | 8.38 | | 160.08 | 9.50 |
| IT | | 773.27 | 17.57 | | 425.11 | 11.63 |
| LT | | 454.57 | 18.67 | | 2317.87 | 44.40 |
| LU | | −10.20 | −10.20 | | | |
| LV | | 364.45 | 1.01 | | 248.66 | 25.30 |
| MT | | 123.15 | 2.60 | | −38.50 | −1.58 |
| NL | | 59.05 | 5.19 | | 42.82 | 4.16 |
| PL | | 239.47 | 22.54 | | 524.40 | 14.15 |
| PT | | 1770.64 | 29.28 | | 290.19 | 5.80 |
| RO | | 1739.51 | 34.55 | | 1905.25 | 29.83 |
| SE | | 149.52 | 10.39 | | 93.10 | 3.75 |
| SI | | 38.22 | 8.34 | | 64.94 | 4.85 |
| SK | | 27108.75 | 2600.39 | | 296511.00 | 39.09 |
| EU | | 123.59 | 5.58 | | 124.46 | 5.59 |

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
