# Peer review of "Analysis and Forecast of the Use of E-Commerce in Enterprises of the European Union States"

_sustainability, doi:10.3390/su14148943_

Round 1

Reviewer 1 Report

The following comments would enhance the reach of the article.

Abstract: EU, DESI and GDP must be written in full form.

Introduction: Mention the literature gap and provide references for the previous contributions towards GDP.

The results need to be compared with previous literature.

What are the basic recommendations from this research?

Author Response

July 15, 2022

The authors are grateful to the Reviewer of manuscript number Sustainability-1814646 for his thoughtful comments.

Response to Reviewer 1 Comments

All of the comments of the reviewer are taken into consideration and answered, point by point, in this letter.

The authors believe that the revised version of the paper is better than the originally submitted one. For this reason, they reiterate their gratitude to the anonymous reviewer.

​In paper revision we used the "Track Changes" function in Microsoft Word.

We would like to thank the Reviewer for kind evaluation of the paper and the positive feedback of our work. The entire article has been revised carefully.

  1. Abstract: EU, DESI and GDP must be written in full form.
  • Authors: We thank the reviewer for the suggestion. Please see the modified Abstract.
  1. Introduction: Mention the literature gap and provide references for the previous contributions towards GDP.”
  • Authors: The authors thank the reviewer for the suggestion. Please see the modified Introduction (lines 66-70).
  1. The results need to be compared with previous literature.”
  • Authors: The authors thank the reviewer for the suggestion. Please see the modified in lines 636-640, 645-665, 669-675.
  1. What are the basic recommendations from this research?
  • Authors: The authors thank the reviewer for the suggestion. Please see the modified in lines 684-695.

Thank you for your thoughtful review. We believe we have responded satisfactorily to your concerns.

Reviewer 2 Report

Line 320: the term 'non-autonomous' is not well known, maybe 'second order autoregressive model with linear trend' would be more clear to the readers.

Line 321: The estimation method is not reported: I suppose that the authors used ordinary least squares.

Figure 1: The bars are not good aesthetically. I would recommend to use points connected with lines. The same problem occurs in Figure 3.

Table 1 and 2: in the caption, clarify that the numbers in the table represent average annual changes.

Figure 2: again, the bars are difficult to read, points connected with lines would be better. I understand that there are many trajectories (one for each country), but, in my opinion, the bars do not allow a proper understanding of the trends. The same problem occurs in Figures 4-8.

Table 3: In the caption, specify the estimation method. Also, 'Prob.' should be 'p-value'.

Figure 10: In the caption, clarify that the blue line represents observed values, while the other lines are fitted values (in 2015-2020) or predictions (2021-2025). Also, explain what 'simple mean' is.

Table 4: 'Regression statistical coefficients' sounds strange to me. Maybe 'Fit statistics' would be a more correct term.

Author Response

July 15, 2022

The authors are grateful to the Reviewer of manuscript number Sustainability-1814646 for his thoughtful comments.

Response to Reviewer 2 Comments

All of the comments of the reviewer are taken into consideration and answered, point by point, in this letter.

The authors believe that the revised version of the paper is better than the originally submitted one. For this reason, they reiterate their gratitude to the anonymous reviewer.

​In paper revision we used the "Track Changes" function in Microsoft Word.

We would like to thank the Reviewer for kind evaluation of the paper and the positive feedback of our work. The entire article has been revised carefully.

  1. Line 320: the term 'non-autonomous' is not well known, maybe 'second order autoregressive model with linear trend' would be more clear to the readers.
  • Authors: We thank the reviewer for the remark. Please see the modified in line 322.
  1. Line 321: The estimation method is not reported: I suppose that the authors used ordinary least squares.”
  • Authors: The authors thank the reviewer for the remark. Please see the modified in line 323.
  1. Figure 1: The bars are not good aesthetically. I would recommend to use points connected with lines. The same problem occurs in Figure 3.”
  • Authors: The authors thank the reviewer for the suggestion. All figures have been modified as suggested. Please see the modified.
  1. Table 1 and 2: in the caption, clarify that the numbers in the table represent average annual changes.
  • Authors: The authors thank the reviewer for the suggestion. The changes were made as your suggested.
  1. Figure 2: again, the bars are difficult to read, points connected with lines would be better. I understand that there are many trajectories (one for each country), but, in my opinion, the bars do not allow a proper understanding of the trends. The same problem occurs in Figures 4-8.”
  • Authors: The authors thank the reviewer for the suggestion. All figures have been modified as suggested. Please see the modified.
  1. Table 3: In the caption, specify the estimation method. Also, 'Prob.' should be 'p-value'.
  • Authors: The authors thank the reviewer for the suggestion. Footnote has been added to the table.
  1. Figure 10: In the caption, clarify that the blue line represents observed values, while the other lines are fitted values (in 2015-2020) or predictions (2021-2025). Also, explain what 'simple mean' is.
  • Authors: The authors thank the reviewer for the suggestion. The changes referring at figure 10 were made as your suggested (lines 553-555). We explain simple mean in lines 555-556.
  1. Table 4: 'Regression statistical coefficients' sounds strange to me. Maybe 'Fit statistics' would be a more correct term.”
  • Authors: The authors thank the reviewer for your remark. The change was made.

Thank you for your thoughtful review. We believe we have responded satisfactorily to your concerns.